# First measurements of tides in the stratosphere and lower mesosphere by ground-based Doppler microwave wind radiometry

Jonas Hagen[1], Klemens Hocke[1], Gunter Stober[1], Simon Pfreundschuh[2], Axel Murk[1], and Niklaus Kämpfer[1]

[1]Institute of Applied Physics, University of Bern, Bern, Switzerland
[2]Department of Space, Earth and Environment, Chalmers University of Technology, Gothenburg, Sweden

**Correspondence:** Jonas Hagen (jonas.hagen@iap.unibe.ch)

**Abstract.**

Atmospheric tides are important for the vertical coupling in the atmosphere from the stratosphere down to the troposphere and up to the thermosphere. They are planetary-scale gravity waves with well-known periods that are integer fractions of a day and can be observed in the temperature or wind field in the atmosphere. Most lidar techniques and satellites measure atmospheric tides only in the temperature field and continuous measurements of the tides in the wind field of the stratosphere and lower mesosphere are rare, even though with modern lidars they would be feasible. In this study, we present measurements of the diurnal tide in the wind field in the stratosphere and lower mesosphere by ground based microwave wind radiometry for two different campaigns in tropical and polar regions. Further, we compare our measurements to MERRA-2 reanalysis data. In the three-monthly mean, we find a good correspondence in the amplitude and phase of the diurnal tide between measurements and reanalysis with the most important features of the diurnal tides represented in both data sets. When looking at shorter timescales, we find significant differences in the data sets. We make an attempt to examine these differences and discriminate between atmospheric variability and noise and present some hints for intermittent diurnal tides. We conclude, that continuous ground based observations of tides in the middle atmospheric wind field are feasible, and deliver consistent results for the mean amplitude and phase of the diurnal tide in the 3-monthly mean. We further discuss the limitations with regards to short timescale observations of tides and the possibility to provide additional insight to middle atmospheric dynamics that is complementary to temperature observations and reanalysis data.

## 1 Introduction

Atmospheric tides are global-scale waves with well-known periods that are integer fractions of a day. They are the result of the periodic solar forcing of the temperature and wind fields and gravity as restoring force. Just as other gravity waves, tidal waves can propagate up or downwards, be reflected and ultimately deposit energy in the atmosphere when they break. This transportation and deposition of energy can cause secondary waves and other disturbances, resulting in a vertical coupling between the horizontal layers of the atmosphere and leads to an exchange of energy and momentum between the forcing regions

and the dissipation altitudes. Ultimately, tides in the stratosphere and mesosphere region can affect weather phenomena like for example the diurnal cycle of tropical rainfall (Woolnough et al., 2004; Sakazaki et al., 2018).

Due to the global nature of atmospheric tides, they have been studied over decades using models (Lindzen, 1971; Forbes and Wu, 2006; Wang et al., 2016) or global observations from satellites (Oberheide et al., 2009; Häusler et al., 2010; Pancheva and Mukhtarov, 2011). Considering the observational results Oberheide et al. (2011) introduced a climatology based model of atmospheric tides covering the most relevant diurnal and semi-diurnal tidal modes at altitudes between $80\,\mathrm{km}$ to $400\,\mathrm{km}$. While atmospheric tides are well understood and modeled (Hagan et al., 1999) on a global and seasonal scale, very little is known about tides on a local and sub-seasonal scale.

Tides in the temperature field have been extracted from satellite observations (Sakazaki et al., 2012; Forbes and Wu, 2006; McLandress et al., 1996; Oberheide et al., 2009) and have been compared to different reanalysis data sets by Sakazaki et al. (2018) from the stratosphere to the lower mesosphere. Satellites, however, often need several weeks to sample a full diurnal cycle for a specific location due to their orbit and therefore are not capable to resolve tidal variations at short timescales. Satellites with a sun-synchronous orbit, like for example Aura MLS, overpass each location on earth at two local times specific to this location and thus sample the diurnal cycle of a specific location with only $12\,\mathrm{h}$ resolution. The global coverage nevertheless enables tidal studies on shorter timescales also for instruments on these satellites (Ortland, 2017).

Ground based measurements of tides in the temperature field have been performed by day-light-capable lidars for the stratosphere by Kopp et al. (2015); Baumgarten and Stober (2019) and from meteor radar temperatures (Stober et al., 2008) in the mesosphere and lower thermosphere (MLT) region. Meteor radar and MF-radar observations are also suitable to obtain tides in the wind fields (Portnyagin et al., 1993, 2004; Merzlyakov et al., 2009; Jacobi, 2012; Wilhelm et al., 2019). Current lidar instruments are able to measure inertial gravity waves in the wind field on short timescales (Baumgarten et al., 2015) and are thus in theory also suited for the observation of atmospheric tides, but the necessity of clear sky conditions reduces the availability of long term observations drastically and no observations of atmospheric tides are available to date.

Rogers et al. (2016) derived the local solar time variation of wind at 95 km altitude by integrating a 5 year data set from different ozone radiometers. Rocket soundings of the tides in the wind and temperature field have been performed by Lindzen and Chapman (1969) up to the upper stratosphere but have never been repeated again.

Note that no observations of tides in the wind field for the stratosphere and lower mesosphere have so far been performed.

This leaves reanalyses data with high temporal resolution like ERA5 and MERRA-2 as the only source for the wind field in studies about atmospheric tides. These products typically depend on satellite measurements and, thus, tides in upper atmospheric region are poorly constrained. Recent findings by Sakazaki et al. (2018) suggest that for the temperature field, differences between the different reanalyses and measurements are systematic in amplitude (approx. $1\,\mathrm{K}$ or $50\,\%$ above $40\,\mathrm{km}$ for northern mid-latitudes, more in tropics) and the spread between the reanalyses is quite large in the lower mesosphere ($0.3\,\mathrm{K}$ to $1\,\mathrm{K}$ at approx. $60\,\mathrm{km}$ for northern mid-latitudes).

Recently, the temporal variability of tides at the MLT as lower forcing of the ionospheric and thermospheric systems become more and more important (e.g., Liu, 2016). There are currently several Global Circulation Models (GCM) developed, which are supposed to describe consistently the vertical coupling between the middle atmosphere and the ionospheric/thermospheric

system (Pancheva et al., 2012; Yiğit et al., 2016; Liu et al., 2018). In particular, the short term variability of the tidal forcing is essential for driving the more complex neutral-ionospheric coupling in the upper atmosphere. McCormack et al. (2017) presented a comparison between a meteorological reanalysis from the Navy Global Environment Model - High Altitude (NAVGEM-HA) and several world wide distributed meteor radars indicating substantial day-to-day variability of the winds and tides. Recently, Baumgarten and Stober (2019) presented a 10-day continuous lidar observation conducted with the Küh- lungsborn Rayleigh-Mie-Raman lidar and estimated the tidal variability using an adaptive spectral filter technique (Stober et al., 2017) and complemented these observations with reanalysis data to investigate the phase relations of temperature and wind tides. However, lidar observations require cloud free conditions, which usually limits the continuity of such time series.

In this study, we use measurements from the ground-based microwave Doppler wind radiometer WIRA-C, that can pro- vide continuous measurements of the wind fields in the stratosphere and lower mesosphere (Hagen et al., 2018). The biggest advantage of the radiometers compared to many other ground based remote sensing instruments is their ability to measure continuously and independent of daylight and light clouds. Further, their compact design makes it rather easy to deploy these instruments at remote locations and enables their autonomous operation. Rüfenacht et al. (2018) performed an initial valida- tion of the technique with other ground based instruments, e.g. the ALOMAR lidar and the Andenes meteor radar. Particularly compared to lidar, radiometers often have a much coarser vertical and temporal resolution. Rüfenacht et al. (2016) examined the spectrum of the wind oscillations for radiometric measurements and model data for periods down to 5 days, which is the lower limit of such an analysis due the low temporal resolution of wind radiometry. In this study, we present a method to inves- tigate sub-day periods of oscillations with microwave wind radiometry by applying a different pre-processing to the measured spectra.

Specifically, we present a method to infer diurnal tides and their variability in the wind field in the stratosphere and lower mesosphere. After a short introduction to the measurement principle and analysis methods, we present the measurements from two WIRA-C campaigns, one was conducted on La Réunion island at tropical latitudes and the other one on Andøya island at polar latitudes. We show that our instrument is able to capture the mean diurnal wind tide over the course of a three-month period, and we compare our measurements to the meteorological reanalysis MERRA-2 (Global Modeling and Assimilation Office (GMAO), 2015) with respect to the amplitude and phase behaviour. We do this comparison for three-monthly means as well on shorter timescales (days/weeks) to reveal some differences between the observations and the reanalysis.

The manuscript is structured as follows. In section 2 we present a summary of the instrument and the campaigns. The data analysis and retrievals are described in section 3. Our results are presented in section 4 and our conclusions are given in section 5.

## 2 The WIRA-C instrument and campaigns

### 2.1 Instrument

The WIRA-C instrument is a Doppler microwave wind radiometer. As described in detail by Hagen et al. (2018), it measures the $142\,GHz$ ozone rotational emission line with a high spectral resolution of $12.5\,kHz$. Because the ozone molecules are moving

with the mean air flow, the Doppler shift introduced to the emission line is directly proportional to the line-of-sight wind speed. In order to be sensitive to the zonal and meridional component of the horizontal wind speed, we observe the emission line for all cardinal directions (North, East, South, West) at a low elevation angle of $22°$. Further, the pressure broadening effect allows the retrieval of altitude resolved wind profiles in an altitude range from $30\,km$ to $75\,km$ on a $3\,km$ vertical grid with $12\,km$ vertical resolution.

WIRA-C has an un-cooled but temperature-stabilized receiver with a low receiver noise temperature due to a state-of-the-art low noise amplifier that directly amplifies the observation frequency of $142\,GHz$. Despite the low noise, integration times of $12\,hours$ to $24\,hours$ are typically applied in the standard retrievals. These long integration times are required to achieve a signal-to-noise ratio that is sufficient for a retrieval of wind speed.

WIRA-C operates autonomously and automatically and the measurements are independent of daylight and light clouds with interruptions only during rain or heavy snowfall. Additionally WIRA-C uses a tipping curve calibration scheme and, thus, only needs very minimal maintenance, most of which can be done remotely. As a result, the WIRA-C instrument is especially well suited for campaigns as well as long-term monitoring observations.

The forward-model for the retrieval is supplied by the ARTS software package (Buehler et al., 2018). The inversion of the measured spectra is performed by an optimal estimation method (OEM) developed by Rodgers (2000). We use the OEM algorithm that has recently been implemented directly into the ARTS software.

Optimal estimation is a method, where the ill-posed inversion problem is regularized by an a priori profile and a corresponding co-variance matrix. It is well suited for the inversion of atmospheric measurements, because the mean background state is often known reasonably well. This applies to this study in particular, where the mean background wind speed is known from models and measurements to a reasonable extent and the diurnal cycle can be understood as a perturbation of the background state that we can retrieve from the measurements.

Different quality control parameters can be derived for an optimal estimation of a profile. The most important to us is the measurement response, that estimates the sensitivity of the retrieved quantity to actual changes in the observed system (as opposed to sensitivity to the a priori profile). Ideally this measurement response is one, with 0.8 or 0.6 being acceptable numbers.

## 2.2 Campaigns

WIRA-C has been on two major campaigns so far. The first campaign started in August 2016 and took place in the southern hemisphere at the Maïdo observatory on La Réunion Island (France) (Baray et al., 2013), located in the Indian ocean at $21\,°S$, $55\,°E$. The Maïdo observatory is located at an altitude of $2200\,m$ a.s.l., which provides ideal conditions for radiometry. At this altitude there is less absorption due to tropospheric water vapor which could be a problem in the tropics at lower altitudes. For tropical latitudes around $\pm30°$, the global scale wave model GSWM (Hagan et al., 1999) predicts a high amplitude of the diurnal tide compared to more polewards or more equatorial latitudes. The campaign ended in January 2018 and we refer to this as the tropical campaign.

For the second (and still ongoing) campaign, WIRA-C was moved to arctic latitudes in June 2018. The instrument is located at the ALOMAR observatory on Andøya (Norway) at 69 °N, 16 °E. We refer to this as the arctic campaign. The ALOMAR observatory is located on mount Ramnan, at 370 m a.s.l. and hosts many other remote sensing instruments e.g., the ALOMAR Rayleigh-Mie-Raman lidar, an Fe-lidar and several radars in the vicinity. The water vapor cycle at ALOMAR is dominated by the tropospheric weather pattern of the marine climate and variable within days rather than within a day.

## 3 Data processing

For the standard WIRA-C time series retrievals as used in previous studies, the spectra are integrated over continuous blocks of 12 or 24 hours resulting in a time series of wind speed with the same resolution (Hagen et al., 2018). Typically, for the tropical site, an integration over 12 hours from sunrise to sunset is performed. However, for the retrieval of the daily cycle we now aggregate the measurements of the same time of day over multiple days to perform a composite analysis. Typically we use a window of 7 to 13 days and aggregate by time of day with a 2 to 4 hour resolution. This gives a total integration time of around 20 hours, centered around a central day. We refer to the different composites by $(\Delta D, \Delta H)$ where $\Delta D$ indicates the number of days and $\Delta H$ the number of hours for the integration. The main composite used in this study is $(13, 2)$, which gives a total integration time of 26 hours.

After integration we run the wind retrieval for the WIRA-C instrument. For MERRA-2 reanalysis data we apply the same composition directly on the model data to get the same temporal smoothing that we have to apply to our measurements. In addition, we also analyse the original reanalysis data.

A major difference to the retrieval described by Hagen et al. (2018) is that we use a non-zero wind a priori profile that corresponds to the mean wind background. This mitigates the effect of the diurnal variability of the troposphere on our measurements. If a zero-wind a priori is used, increased noise during daytime could lead to an over-estimation of tidal amplitudes in the subsequent analysis, because the retrieved wind speed would be closer to zero (and thus possibly further away from the background) in case of increased noise during daytime. In contrast, a mean-background a priori in combination with poor measurement response would lead to an underestimation of tidal amplitudes. This is especially important for locations with high diurnal variability (like our tropical site) or frequent rainfall (like our arctic site close to the sea). We extract the mean wind background from ECMWF operational data and average over the full 13 days centered around the (13,2) composite and analogous for the other composites. Like this, our a priori wind profile does not include any tidal information at all. For the ozone a priori data, we compose WACCM data from Schanz et al. (2014) analogous to our measurements.

Once we have retrieved the wind profiles (or extracted them from the reanalysis data), we fit a simple tidal model to extract amplitude and phase information. The simple model for an arbitrary observable quantity $y$ has the form of

$$y(t) = c + \sum_{k=1...N} A_k \cos(t\frac{2\pi}{P_k} - \phi_k) \tag{1}$$

$$= c + \sum_{k=1..N} \left[ a_k \cos(t\frac{2\pi}{P_k}) + b_k \sin(t\frac{2\pi}{P_k}) \right] \tag{2}$$

where $P_k = 24, 12, 8, \ldots$ h is the period of the diurnal, semi-diurnal and ter-diurnal tide. In this study we use $N = 1$ and only consider the diurnal tide, but we write down the full basis in (2) to point out that the components for $k = 1, 2, 3, \ldots$ are orthogonal and can thus be treated separately. We apply a least squares optimization on (2) for the zonal and meridional wind component and assume the same weight for all wind measurements. Additionally, we estimate the uncertainty of the fit from the error co-variance matrix of the adjusted parameters.

Equation (1) defines the phase $\phi_k$ as the time of day when the corresponding wind component has its maximum. Note, that we present phases in units of mean solar time, so for example a phase of $10\,\mathrm{h}$ means that the maximum occurs $2\,\mathrm{h}$ before noon of the mean solar day. The amplitude $A_1$ and phase $\phi_1$ of the diurnal tide are finally given by

$$A_1 = \sqrt{a_1^2 + b_1^2} \tag{3}$$

$$\phi_1 = \arctan2\,(b_1, a_1) \ \in (-\pi, \pi]. \tag{4}$$

Since we average over multiple days prior to the retrieval, we do not apply a windowing function for the fitting of the tide as it is often suggested to compensate for the intermittency of waves. We assume that the retrieval of averaged spectra yields the average wind speed, so windowing and aggregation can be considered equivalent. This assumption might not hold in the context of fast changes in the wind field and we thus prefer periods of a stable wind background for our detailed analysis. Especially we do not attempt to retrieve tidal parameters during strong planetary wave activity, nor in the context of extreme events like sudden stratospheric warmings. Further, we consider non-tidal gravity waves to be filtered out by the vertical smoothing of the instrument of about $12\,\mathrm{km}$.

Vertical smoothing (artificially or due to instrumental properties) decreases tidal amplitudes depending on the vertical wavelength of the observed tides. If the vertical wavelength is infinite (tidal phase is constant with altitude), the amplitude is not affected, whereas at typical vertical wavelengths of the diurnal tide of around $30\,\mathrm{km}$, the smoothing can reduce the observed amplitude by up to $0.25$. In this study, we do not apply any vertical smoothing to the reanalysis data.

To check for the significance and robustness of the diurnal tidal parameters, we compare the outcome for different composites. We run the same analysis for the (13,2), (11,2), (9,3), (9,2), (7,4) and (7,3) composites, which provide different samplings of the same observable. The similarities and differences among all the composites indicate how robust the parameters are and allow us to estimate the influence of noise from instrumental and atmospheric sources in a qualitative way.

We also compute the mean amplitude and phase over a larger time span (three months) by averaging the wind field prior to fitting the tide-model. We estimate the uncertainty of the amplitude and phase for the 3-monthly mean using a bootstrapping method that follows the Moving Block Bootstrap (Lahiri, 2003, p.25ff). For a $(\Delta D, \Delta H)$ composite time series, we sample a synthetic three-month period (91 days) by choosing $\frac{91}{\Delta D}$ composite days at random and estimate the diurnal tidal parameters of the mean diurnal cycle for each sample. The distribution of the parameters gives us an estimate on the uncertainty due to observational errors as well as due to phase variability during the period of observation.

## 4 Results

From both campaigns we select a three months period, based on the following criteria. First we look for periods with stable background wind conditions. Due to the composition (maximum 13 day) for the wind retrieval, we prefer time intervals with no extreme meteorological events, e.g. sudden stratospheric warmings (SSW), and a low planetary wave activity because this might impact the retrieval of tidal amplitudes.

Further, we only considered time intervals with a good overall measurement response, which corresponds mainly to little precipitation. Another important aspect for the data analysis is the continuity of the observations (minimal instrumental downtime) to avoid issues in compiling the temporal averages.

Considering the above mentioned criteria we decided to focus on two campaign intervals from April to June 2017 at Maïdo and from September to November 2018 at Andenes (ALOMAR).

This gives us two time-series, one for the tropical campaign and one for the arctic campaign, for which we perform the previously described analysis. We use local mean solar time for all plots and phases in this study, which is simply a fixed offset depending on longitude. For the tropical campaign at $55.5°$ longitude, this is an offset to UT of $3.7\,\mathrm{hours}$. For the arctic campaign at $15.7°$ longitude, the offset of local mean solar time to UT is $1\,\mathrm{hour}$.

### 4.1 Results for the tropical campaign

For the tropical campaign, we selected the period from April to June 2017. This period is at the beginning of austral winter where reanalysis as well as measurements show a steady background of strong eastward winds with a small meridional component and relatively low planetary wave activity. Moreover, during this rather dry season, WIRA-C performed well and measured continuously with a good measurement response.

The meteorological background wind field for the selected period at the Maïdo observatory is shown in Fig. 1 as measured by WIRA-C, complemented with MERRA-2 reanalysis for lower altitudes. The zonal and meridional winds indicate some variability at temporal scales of a few days. Characteristic for the zonal winds are westward winds below $40\,\mathrm{km}$ altitude and a strong zonal eastward stratospheric jet from $45\,\mathrm{km}$ to $70\,\mathrm{km}$ altitude, which intensifies at the beginning of May. Meridional winds exhibit a steady change between southward and northward winds within a few days. Corresponding to the zonal wind enhancement, meridional winds become more southward at beginning of May above a height of $60\,\mathrm{km}$.

The average diurnal cycle on different altitudes over the whole three month period is shown in Fig. 2 for WIRA-C measurements and MERRA-2 reanalysis. In both data sets, the diurnal tide is readily visible. The reanalysis data seems not to contain any other modes than the diurnal tide and the tide model from Eq. (2) fully fits the data. The measurements expose some more variability, especially at higher altitudes and during the afternoon hours. This is related to increased noise in the measurement in the afternoon hours, which is most prominent for the westwards observation direction (and thus only seen in zonal wind retrievals) due to local weather patterns at the Maïdo observatory on La Réunion Island.

The agreement between MERRA-2 and the WIRA-C wind retrievals with respect to the mean behaviour can be assessed in Fig. 3. The left panel (Fig. 3a) shows the amplitude of the zonal and meridional mean diurnal cycle over the entire campaign

period for WIRA-C and MERRA-2. Both data sets show a similar profile with relatively low amplitudes of less than $5\,\mathrm{m\,s^{-1}}$ below $55\,\mathrm{km}$ and slightly higher amplitudes for the meridional component. Figure 3b shows the same for the vertical phase behaviour. The profiles of measurements and reanalysis are in agreement with respect to the amplitudes and phases up to an altitude of $55\,\mathrm{km}$ for the tropical location where the reanalysis data lies within or is close to the limits of confidence of our measurements. The amplitude of the diurnal tide agrees for the measurement and reanalysis within or close to their limits of confidence. Above $55\,\mathrm{km}$ altitude, an increased discrepancy is evident for the meridional wind between MERRA-2 and the radiometer.

The phase of the mean daily cycle measured by WIRA-C as shown in Fig. 3b indicates a vertical wavelength of about $30\,\mathrm{km}$. Approximately the same vertical wavelength is found in the reanalysis. Above $55\,\mathrm{km}$ altitude, the vertical wavelength of our measurements increases drastically and the phase eventually becomes constant with altitude. Evidently, the tide seen by WIRA-C lags behind the tide represented in the reanalysis by $5\,\mathrm{hours}$ at the lower most altitude levels. Currently, we cannot explain this offset.

Figure 9a shows the phase difference between zonal and meridional diurnal tide. Measurements and reanalysis show a remarkable agreement and show that the meridional tide leads the zonal tide by approximately $6\,\mathrm{hours}$ ($90°$ phase angle) as is expected for the southern hemisphere.

Besides the tri-monthly mean, we show the diurnal tidal amplitude $A_1$ and phase $\Phi_1$ versus time for the whole period in Fig. 4. The upper two panels (Fig. 4a and 4b) show the WIRA-C retrievals. The central two panels (Fig. 4c and 4d) are obtained from the composed MERRA-2 reanalysis, whereas the bottom two panels (Fig. 4e and 4f) show the data for the original MERRA-2 reanalysis. These figures present the outcome for the (13, 2) composite only and we provide five more composites in the appendix in Fig. A1 and Fig. A2 for the tropical campaign. The WIRA-C diurnal tidal amplitudes reach a maximum at the beginning of the campaign of about $16\,\mathrm{m\,s^{-1}}$ to $20\,\mathrm{m\,s^{-1}}$ for both components. During the second phase of the campaign we observe smaller tidal amplitudes of approximately $4\,\mathrm{m\,s^{-1}}$ to $12\,\mathrm{m\,s^{-1}}$. The vertical structure that is obvious in Fig. 4a can also be seen in the three-monthly mean (Fig. 3a) with a minimum at $55\,\mathrm{km}$ in reanalysis and measurements. A possible source of this structure is the mixing of different tidal waves with different vertical wavelengths or propagation directions.

Further, our observations show a strong time dependence of the diurnal tidal amplitude and phase at the resolved temporal scales of 7 to 13 days. We observe the same morphology in the time series for all composites (Fig. A1 and Fig. A2) and take this as a hint that the variability is not only due to noise. Nevertheless, there are some differences between the different composites, especially in the absolute values of the amplitudes that are most probably related to instrumental noise. The phase on the other hand is more robust and the time series of the diurnal tidal phase has the same structure in all composites and exposes a pattern of the diurnal tidal phase modulated with a period of approximately a month.

Compared to the WIRA-C observations, the MERRA-2 reanalysis data shows a constant amplitude and phase over time in the composite analysis (Fig. 4c and 4d). The original MERRA-2 data as shown in Fig. 4c and 4d is less constant over time and phase and amplitude expose a high day-to-day variability. However, the mean behaviour seems to be in good agreement to the

observations and the general morphology of the diurnal tide in amplitude and phase seems to agree between measurements and reanalysis.

Figure 5 shows a comparison of the time series between WIRA-C and the MERRA-2 reanalysis at an altitude of $55\,\text{km}$. The shaded area represents the uncertainties of the estimated diurnal tidal amplitudes taken from the co-variance of the adjusted parameters. In addition to the MERRA-2 reanalysis that has been smoothed with our integration kernel, we show the tidal analysis of the original MERRA-2 data. This comparison indicates how the diurnal tidal amplitude decreases and increases again during the campaign in the measurements as well as in the reanalysis. MERRA-2 and our measurements show larger diurnal tidal amplitudes at the beginning of the tropical campaign in April, which then decrease in May and June.

## 4.2 Results for the arctic campaign

From the ongoing arctic campaign in Andenes (ALOMAR), we selected September, October and November 2018 for our analysis. The measurement response during these months is between 0.8 and 1.2 between 42 and 62 km and between 0.7 and 1.3 between 39 and 69 km and therefore acceptable for the whole altitude range we cover. We did not choose this period to start earlier because of the biannual wind reversal that took place just before September 2018 and, on the other end, did not expand this period to December because of the major sudden stratospheric warming that took place in this winter (Schranz et al., 2019).

For this period, the meteorological situation is dominated by the fall transition in the stratosphere. The corresponding background wind field retrieval from WIRA-C is shown in Fig. 6. The campaign period starts at the end of summer with a weak eastward zonal jet between $45\,\text{km}$ to $55\,\text{km}$ altitude, which evolves into a typical polar vortex until November covering nearly all observed altitudes from $40\,\text{km}$ to $70\,\text{km}$. Meridional winds are dominated by a southward flow at the beginning of the campaign period, which then reverses into a northward wind regime at the end of October. Both wind components indicate some variability due to waves on temporal scales of a few days, in particular, the meridional wind indicates an onset of the planetary wave activity towards the end of the observation period.

Figure 7 shows the diurnal cycle, averaged over the three months during the arctic campaign. The reanalysis data is well fitted by our diurnal tide model on all altitudes, but presence of a semi-diurnal component is indicated by the slight oscillation of the reanalysis data around the diurnal tide model fit. The WIRA-C measurements are well represented by the simple diurnal tide model, while containing some higher frequency oscillations that could originate in higher mode oscillation as well as measurement noise.

Figure 8 shows amplitude and phase profiles of the mean daily cycle of this period. Measurements and reanalysis mostly agree and show the same structure of the diurnal tidal amplitude and phase. Still as seen in Fig. 8a, WIRA-C measures a higher diurnal amplitude than the reanalysis suggests. Here, the offset in amplitude seems to be systematic since for the most part, the profiles do not agree within their limits of confidence even though they show the same structure.

The phase of the mean daily cycle is shown in Fig. 8b. Both data sets show a situation with infinite vertical wavelength above $45\,\text{km}$. The maximum of the diurnal tide is around noon and 18h local time for the zonal and meridional component respectively.

The phase difference between zonal and meridional component in the three-monthly mean is shown in Fig. 9b. Above $45\,\mathrm{km}$, the zonal tide leads the meridional tide by $6\,\mathrm{hours}$ ($90°$ phase angle), as expected for the northern hemisphere. Below $45\,\mathrm{km}$, the phase of the zonal component starts to deviate in the measurements and gets behind the meridional tide at approximately $40\,\mathrm{km}$ altitude.

Figure 10 shows extracted amplitudes and phases of the diurnal tide for the arctic campaign over time and altitude. The panels on the left (Fig. 10a, 10c and 10e)) show the diurnal tidal amplitude $A_1$ for the WIRA-C measurements and the MERRA-2 composite and original reanalysis respectively. In the measurements, the diurnal tide is stronger in September than it is in October and reaches nearly $20\,\mathrm{m\,s^{-1}}$ in the meridional component by end of September. Again, this is most probably related to augmentation of tides by weaker background wind speed caused by the seasonal wind reversal that took place at the end

of August to beginning of September 2018. But also during October and November, diurnal tidal amplitudes are quite strong, often up to or more than $15\,\mathrm{m\,s^{-1}}$. This is different for the composite reanalysis data, where amplitudes are generally lower than $10\,\mathrm{m\,s^{-1}}$. In the non-smoothed reanalysis data, tidal amplitudes of more than $15\,\mathrm{m\,s^{-1}}$ are also reached.

     The extracted diurnal tidal phase $\Phi_1$ is shown in Fig. 10b and Fig. 10d for the measurement and reanalysis respectively. In general, the reanalysis shows a very stable phase with the maximum of the diurnal tide at noon local time for the zonal

component and 18h local time for the meridional component. Only in November, the phase becomes more variable and exposes some structure which might be related to the onset of planetary wave activity. This applies to the smoothed and orignal MERRA-2 data equally and is in strong contrast to the measurements, where the diurnal tidal phase is highly variable with time and altitude on the presented timescales. Because this time-dependence is present in all composites with the same morphology, we partly attribute it to the intermittency of the diurnal tide while the exact value of the phase might still be subject to noise.

As for the tropical campaign, we discuss the (13,2) composite and provide a total of six composites in the appendix (Fig. A3 and Fig. A4). Again we see the same morphology and time-dependence of amplitude and phase in all composites. While the amplitude is less consistent among the composites due to noise, the phase is more robust.

     Figure 11 shows the time series for one specific altitude level ($53\,\mathrm{km}$) and additionally shows the amplitudes of the original reanalysis data, that has not been composed over multiple days. Especially in November, the original reanalysis data shows a

25 variability that is comparable or even stronger compared to the measurements. By applying the composition, this variability is averaged out from the reanalysis data but not entirely from the WIRA-C measurements. This indicates that the coherence time of short timescale disturbances of the diurnal tide might actually be longer than the reanalysis predicts. Notably, the measured amplitude features a disturbance around September 24th, where the zonal amplitude is close to zero and the meridional amplitude exhibits a maximum. Similar dynamics are not represented in the reanalysis data, not even in the (not-smoothed) original

reanalysis data.

## 4.3   Summary

We presented measurements of the diurnal tide in the wind field in the stratosphere and lower mesosphere. To our knowledge, these are the first direct observations of tides in the wind field in the middle atmosphere ($30\,\mathrm{km}$ to $70\,\mathrm{km}$) with ground based instruments. In contrast to the standard time series retrievals applied in previous studies, we apply a composite analysis and

superpose spectra for the same time-of-day over several days. This composite analysis enables us to resolve tidal structures in the wind field.

To investigate the results of our method, we applied our analysis to two three-month periods from different campaigns, one from the southern hemisphere and one from the northern hemisphere.

The averaged data over the three month periods showed basic and well-known properties of the diurnal tide with different details for the northern and southern hemisphere. Notably we observed an increasing amplitude with altitude, reasonable vertical wavelengths and a $6\,\mathrm{hours}$ ($90°$ phase angle) shift between zonal and meridional component with the leading component being different for the the austral and boreal locations.

We compared our wind measurements to the MERRA-2 reanalysis which has already been compared by Sakazaki et al. (2018) to other data sets with regard to tides in the temperature field. We find a good over-all-correspondence between reanalysis and measurements in the amplitude of the diurnal tide in the wind field and temporal evolution thereof. Further the amplitude and phase profiles for the 3-monthly mean wind field correspond between the radiometer and the meteorological reanalysis, mostly within their uncertainties for the tropical campaign and with a small offset for the arctic campaign. While the phase of the diurnal tide is very stable in the reanalysis data, especially in the polar region and sometimes even stationary, we see a big temporal variation of the phase in our measurements which persists among different composites.

Further, we presented a time-series of the diurnal tidal amplitude and phase over two three-month periods. We observed an augmented diurnal tide in the context of weak background wind speeds after seasonal wind reversals. Looking at shorter timescales, we observe strong temporal variability of the amplitudes and especially of the phase. Certainly, the time dependence of the diurnal tidal phase has to be investigated in further studies. Since we see the same time dependence in all our composites, we conclude that intermittent diurnal tides could be a possible explanation.

## 5 Conclusions

In summary, we find that reanalysis and measurements agree on the tidal component in the 3-monthly mean daily cycle. We conclude, that the MERRA-2 reanalysis captures the amplitude and phase of the mean diurnal tide reasonably well when averaging over 3 months or longer. When looking at shorter timescales, obvious differences between model and measurements appear. We can explain some of the differences like the augmented tide in context of weak background winds and also observe other notable differences like the variable phase in measurements that are not represented in the MERRA-2 product. Since the general morphology of the variable phase is the same among all composites, we conclude that it might be caused by actual atmospheric variability of the tide. While these intermittent structures are present to some extent in the reanalysis data as well, they are averaged out when we apply the 13-day smoothing that is equivalent to the composite of our measurement. A possible explanation would be, that the coherence time of short time scale disturbances of the diurnal tide is longer and variability is in general stronger in reality than in the reanalysis model. On the other hand, our study is based on a composite analysis, and we assume that the diurnal tide is reasonably stationary during our 7 or 13-day window. Baumgarten and Stober (2019), based

on temperature lidar observations with high temporal resolution, provide some evidence that tides can be highly variable and composition windows should be no longer than a few days.

Further studies could develop more elaborate methods to extract tidal information from radiometer data and further constrain the uncertainty on the extracted parameters. Diurnal tide measurements from daylight-capable lidar or even from rocket campaigns would be a possible source of comparison data. Also other models, for example NAVGEM-HA, could provide further comparison data on different time-scales.

In this study, we focused on the diurnal tidal component only. Future studies could as well address the semi-diurnal component using the same or different composites.

We required a stationary background and focused on selected periods which satisfy this condition. Further studies could investigate the necessity or this requirement and retrieve tides while taking a variable background into account.

Ground-based microwave radiometers are capable to measure continuously over decades and can be deployed in tropical and polar latitudes with minimal maintenance and deployment effort. This makes these instruments very valuable to observe dynamics in the atmosphere, especially from $30\,\mathrm{km}$ to $70\,\mathrm{km}$ altitude where observations are scarce. Since tidal waves on sub-seasonal and regional scales play an important role in the dynamics of the atmosphere, such observations are highly valuable.

*Data availability.* The retrieved wind fields (WIRA-C level 2 data) for the two 3-month periods are available from the zenodo public repository (doi:10.5281/zenodo.3468900).

*Author contributions.* JH performed the data analysis and wrote the manuscript. KH and GS helped to interpret the results and contributed to the writing. SP contributes to the ARTS software package and worked on retrieval code. AM contributed to the development of the instrument
and helped to discuss the method and the results. NK is PI of the project.

*Competing interests.* No competing interests have been identified by the authors.

*Acknowledgements.* The authors the staff of the ALOMAR observatory on Andøya, Norway and the collegues from the IAP Kühlungsborn, Germany for their support and help during the campaign. The authors acknowledge the European Communities, the Région Réunion, CNRS, and Université de la Réunion for their support and contributions in the construction phase of the research infrastructure OPAR (Observatoire
de Physique de l'Atmosphère de La Réunion). OPAR is presently funded by CNRS (INSU), Météo France and Université de La Réunion and managed by OSU-R (Observatoire des Sciences de l'Univers de La Réunion, UMS 3365). The authors acknowledge the European Centre for Medium-Range Weather Forecasts (ECMWF) for the supplied data. This project has received funding from the European Union's Horizon 2020 Research and Innovation program under grant agreement no. 653980 (ARISE2) and was supported by the Swiss National Science Foundation (SNF) under grant number 200020-160048 and the Swiss State Secretariat for Education, Research and Innovation
(SBFI) contract 15.0262/REF-1131-/52107.

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

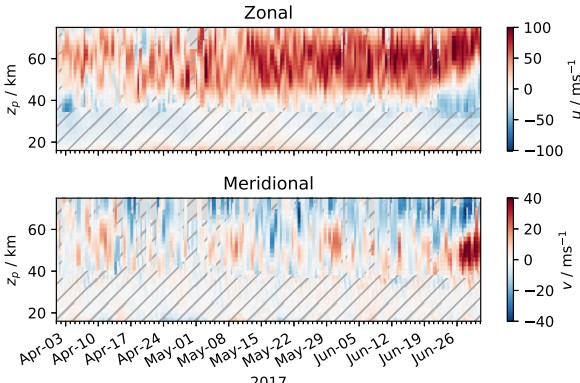

**Figure 1.** Background wind speed measured by WIRA-C complemented with MERRA-2 reanalysis data (hatched area) for the tropical campaign. Note the different scales for the zonal and meridional component.

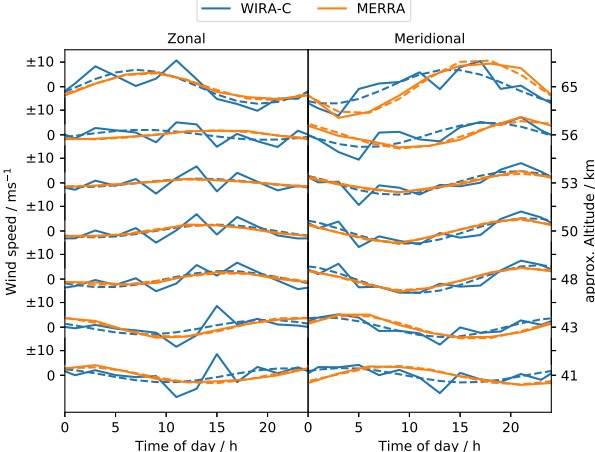

**Figure 2.** Mean daily cycle of zonal and meridional wind speeds in different altitudes for the three months during the tropical campaign from MERRA-2 and WIRA-C. Dashed lines indicate the best fit of the diurnal tide model.

(a) Amplitude $A_1$        (b) Phase $\Phi_1$

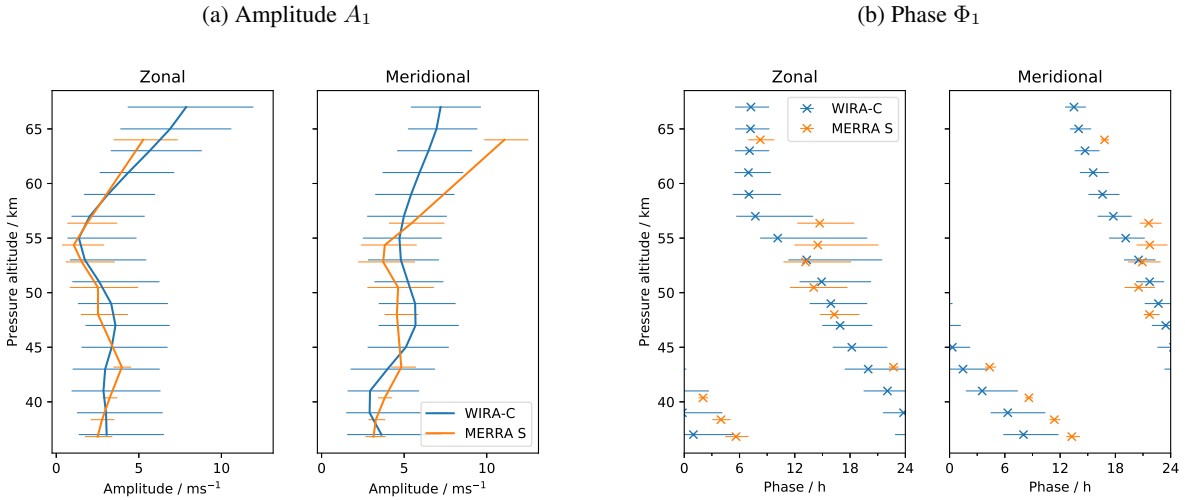

**Figure 3.** Amplitude and phase of mean daily cycle over three months from MERRA-2 reanalysis and WIRA-C measurements from the tropical campaign. Error bars indicate 95% confidence limits. Phase is equivalent to solar time of maximum.

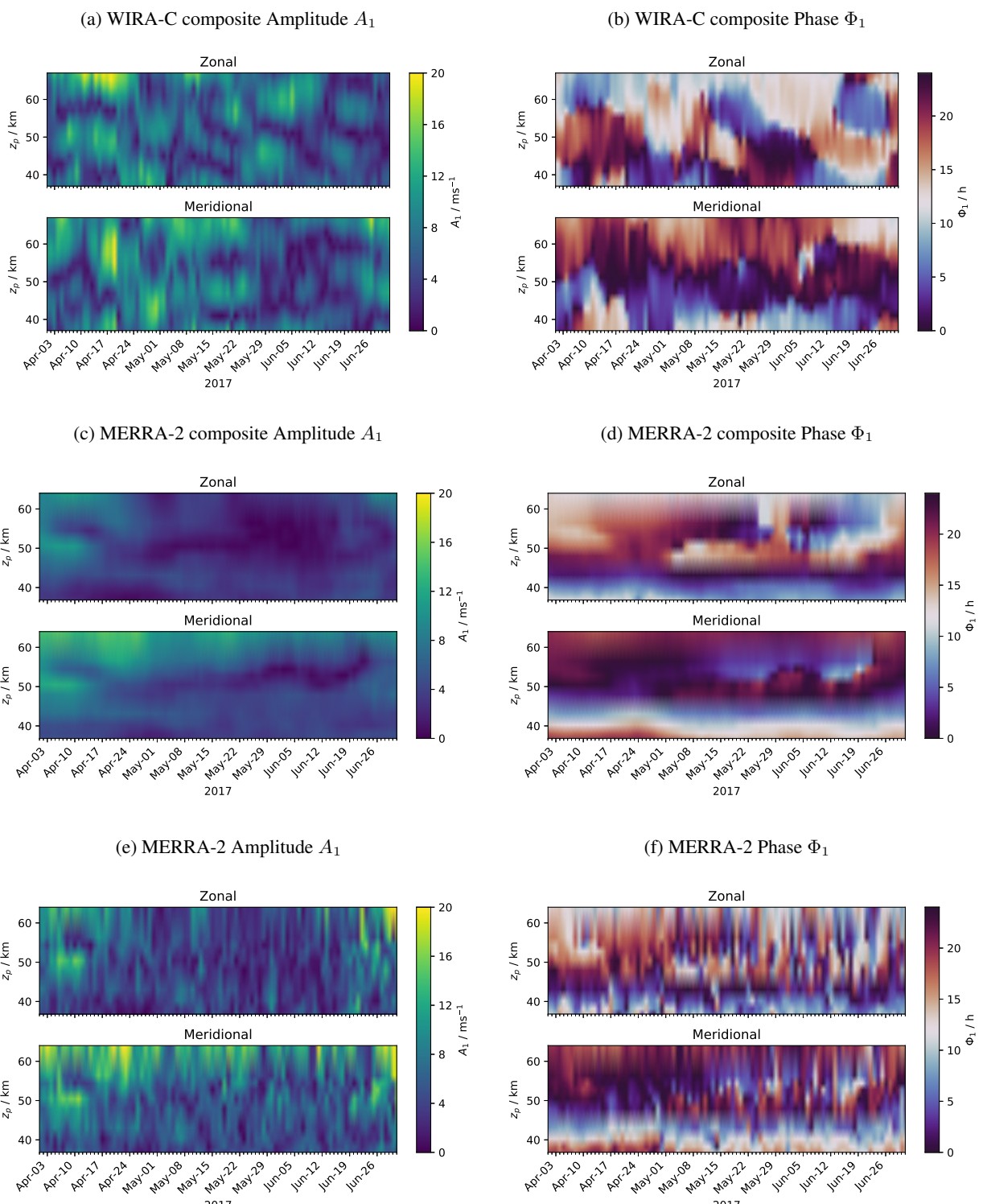

**Figure 4.** Amplitude and phase of the diurnal tide over three months from WIRA-C measurements (a,b), smoothed MERRA-2 reanalysis (c,d), and original MERRA-2 (e,f) during the tropical campaign at the Maïdo observatory.

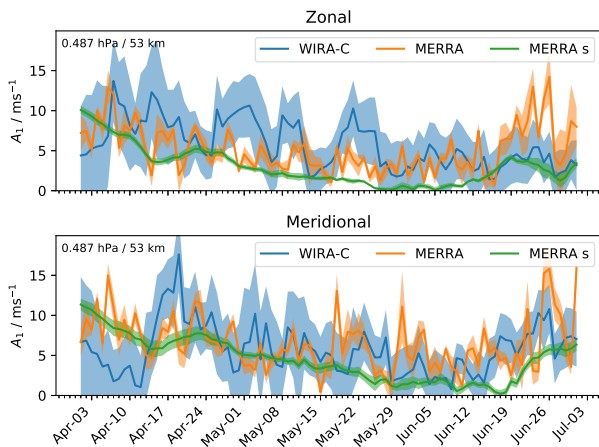

**Figure 5.** Time series of zonal and meridional diurnal tide amplitudes during the tropical campaign for measurements (WIRA-C), MERRA-2 reanalysis (MERRA) and reanalysis smoothed (MERRA s) at an altitude level of 53 km with corresponding errors of the model fit as shaded area.

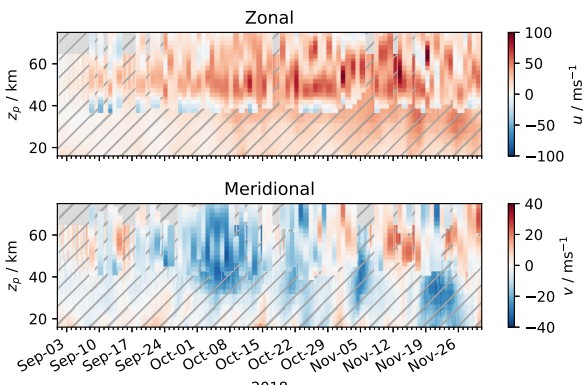

**Figure 6.** Background wind speed measured by WIRA-C complemented with MERRA-2 reanalysis data (hatched area) for the arctic campaign. Note the different scales for the zonal and meridional component.

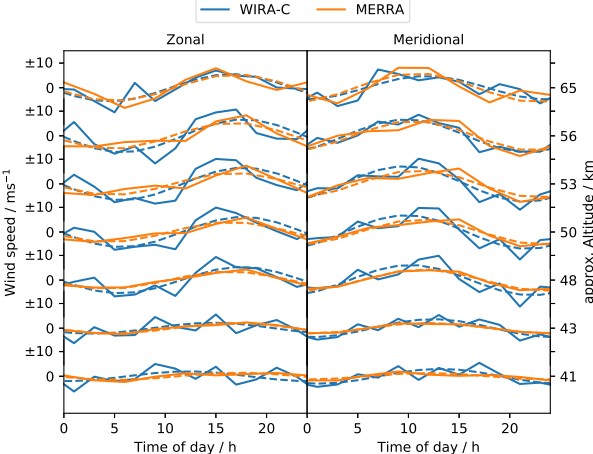

**Figure 7.** Mean daily cycle of zonal and meridional wind speeds in different altitudes for the three months during the arctic campaign from MERRA-2 and WIRA-C. Dashed lines indicate the best fit of the diurnal tide model.

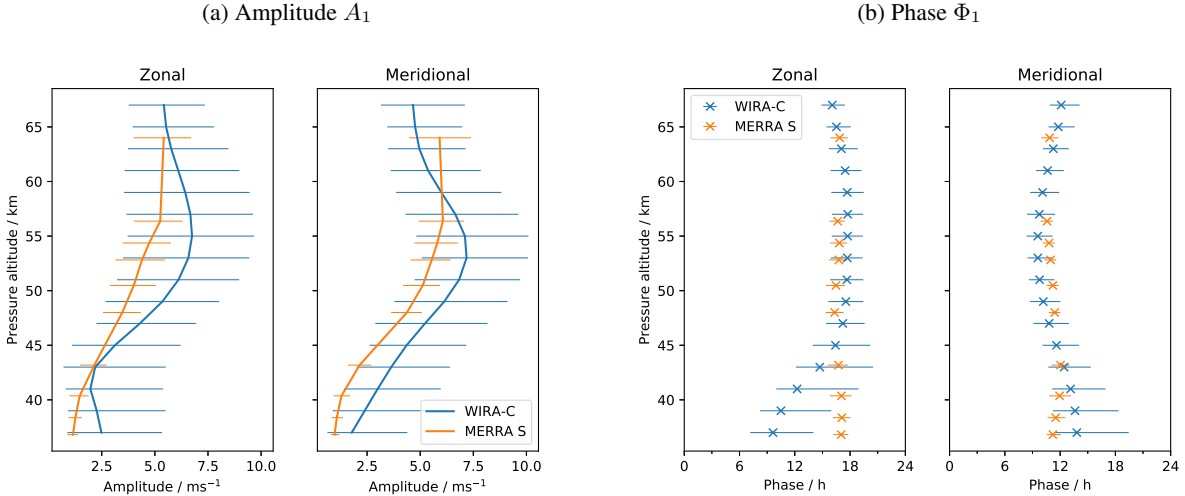

**Figure 8.** Amplitude and phase of mean daily cycle over three months from MERRA-2 reanalysis and WIRA-C measurements from the arctic campaign. Error bars indicate 95% confidence limits. Phase is equivalent to solar time of maximum.

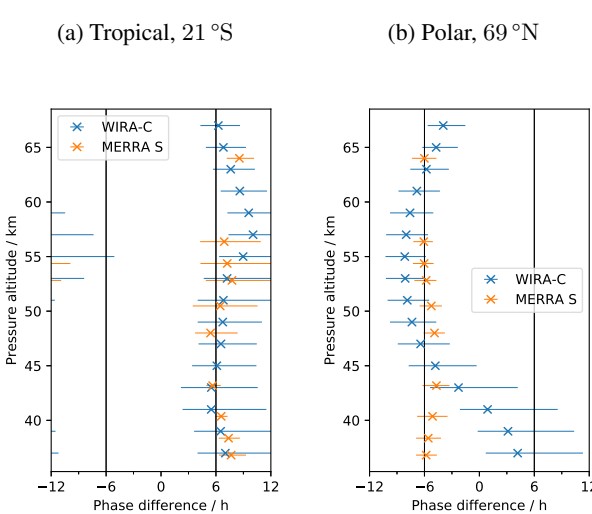

**Figure 9.** Phase difference between meridional and zonal diurnal wind tide ($\Delta\phi = \phi_{\mathrm{merid}} - \phi_{\mathrm{zonal}}$) for the tropcal and arctic campaigns in WIRA-C measurements and MERRA-2 reanalysis.

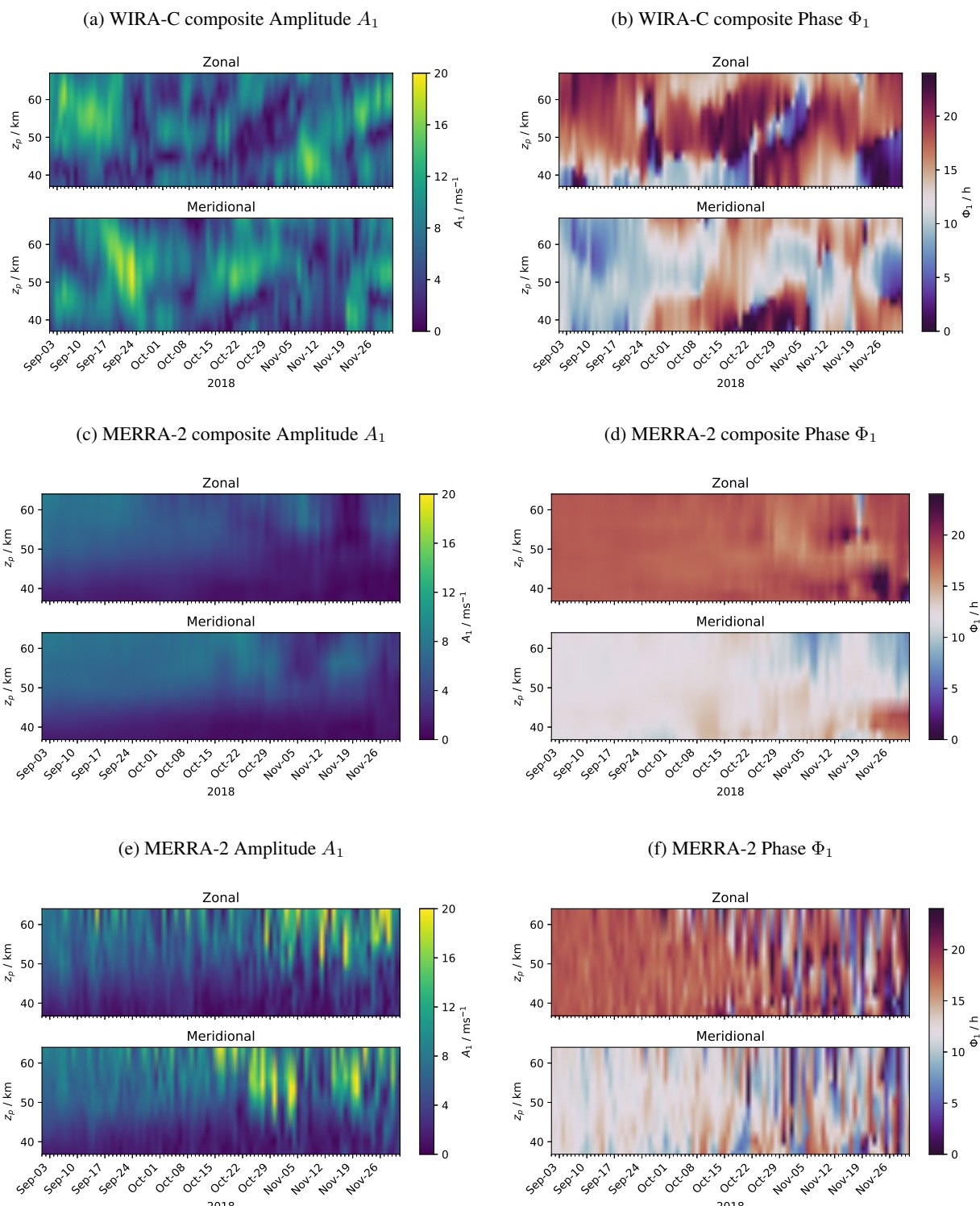

**Figure 10.** Amplitude and phase of the diurnal tide over three months from WIRA-C measurements (a,b), smoothed MERRA-2 reanalysis (c,d), and original MERRA-2 (e,f) during the arctic campaign at the ALOMAR observatory.

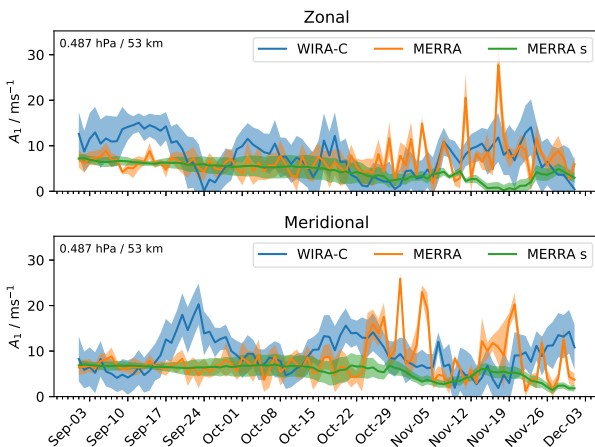

**Figure 11.** Time series of zonal and meridional diurnal tide amplitudes during the arctic campaign for measurements (WIRA-C), MERRA-2 reanalysis (MERRA) and reanalysis smoothed (MERRA s) at an altitude level of 53 km with corresponding errors of the model fit as shaded area.

## Appendix A: Diurnal tide amplitude and phase for all composites

Figures A1, A2, A3, A4 contain panels with the extracted amplitude and phase of the diurnal tide in the wind field for six different composites of the WIRA-C wind measurements. We refer to the different composites by $(\Delta D, \Delta H)$ where $\Delta D$ indicates the number of days and $\Delta H$ the number of hours for the integration, the total integration time is thus given by $\Delta D \times \Delta H$ and ranges from $21\,\mathrm{h}$ to $28\,\mathrm{h}$ for the presented composites. We select these six composites because they all have an integration time of around $24\,\mathrm{h}$ and provide different window lengths (7 to 13 days) and different resolutions (2 to 4 hours).

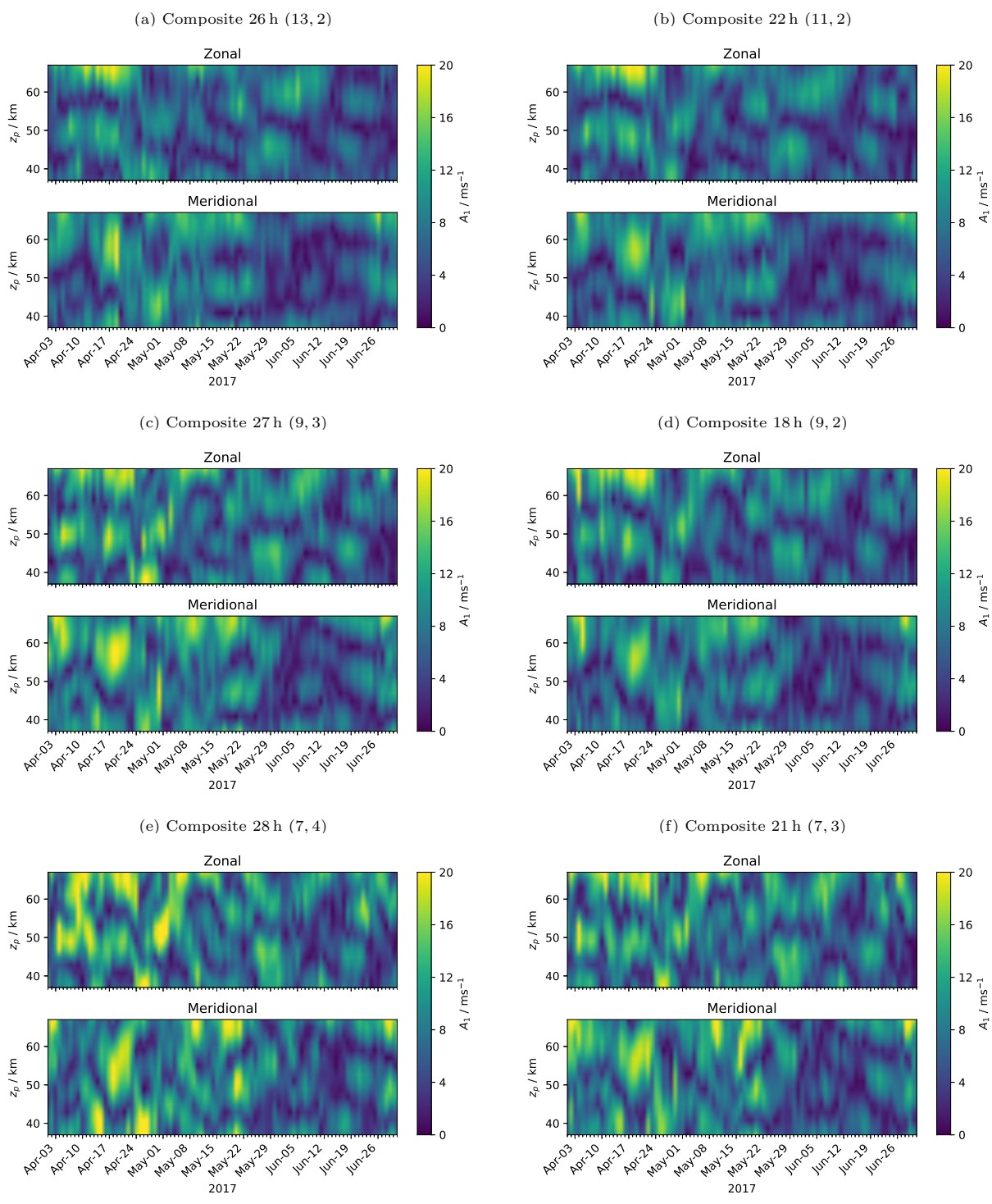

**Figure A1.** Tropical campaign: Amplitude of the diurnal tide for six different composites with a similar total integration time of 21 h to 28 h ordered by window length.

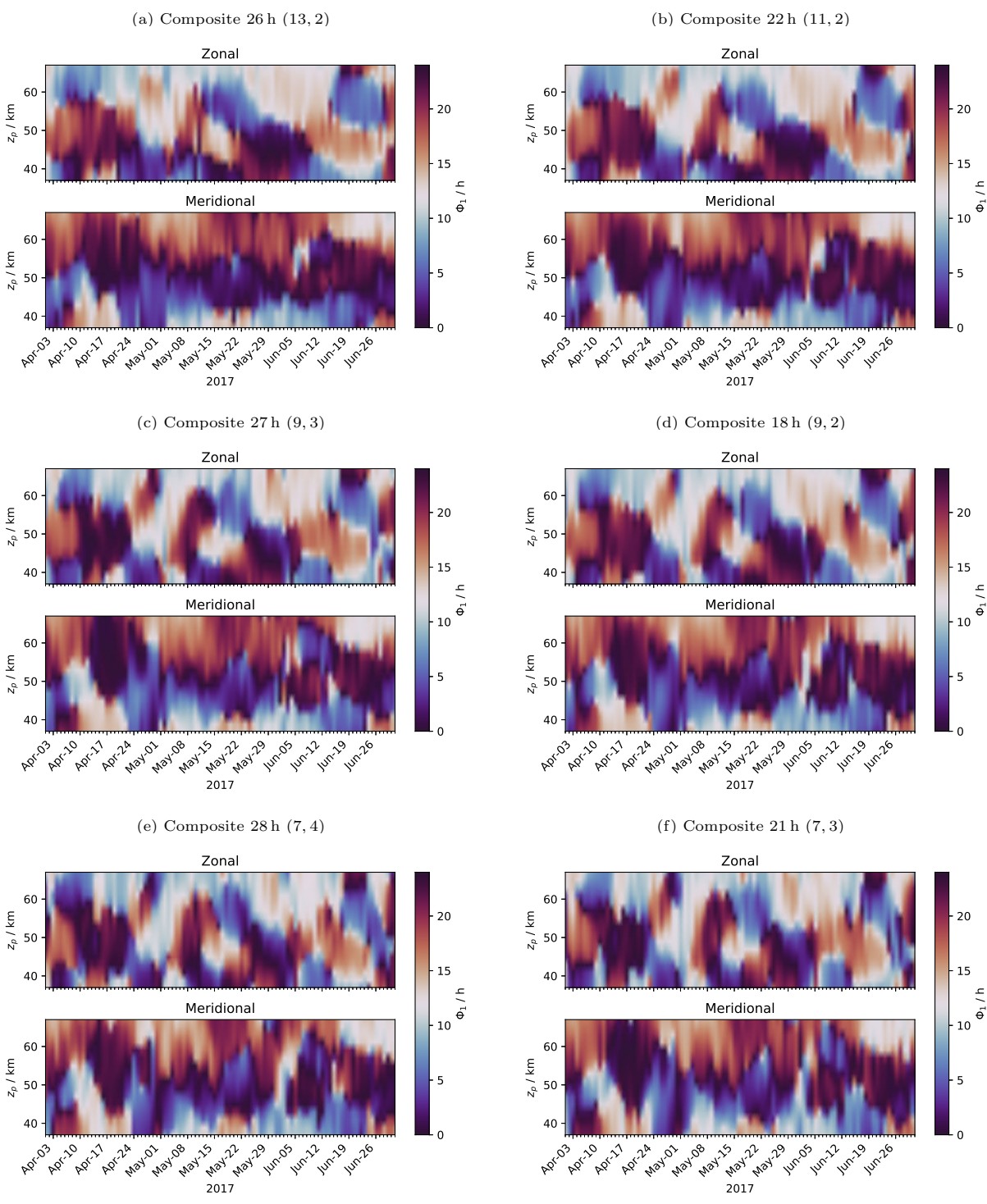

**Figure A2.** Tropical campaign: Phase of the diurnal tide for six different composites with a similar total integration time of 21 h to 28 h ordered by window length.

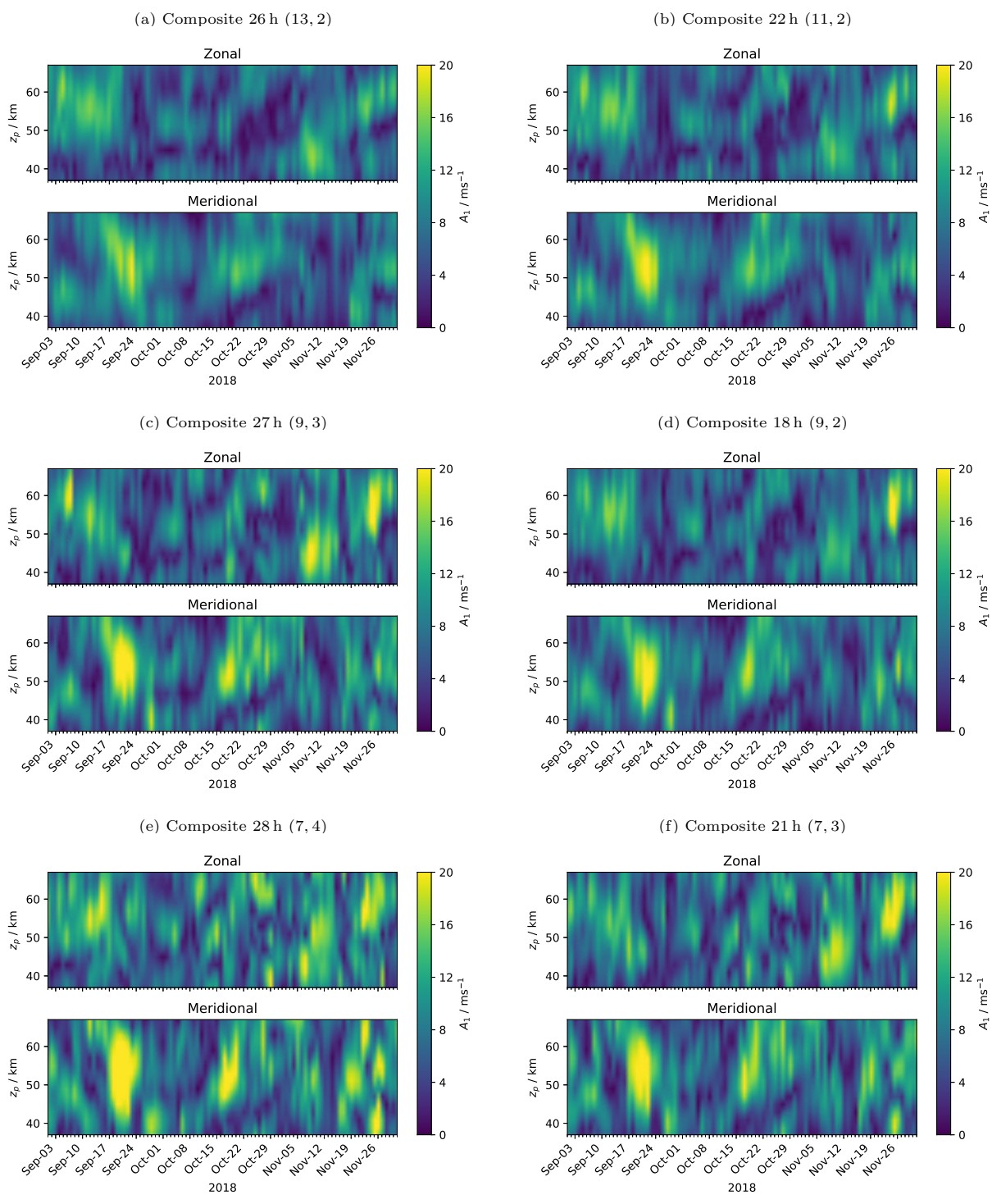

**Figure A3.** Arctic campaign: Amplitude of the diurnal tide for six different composites with a similar total integration time of 21 h to 28 h ordered by window length.

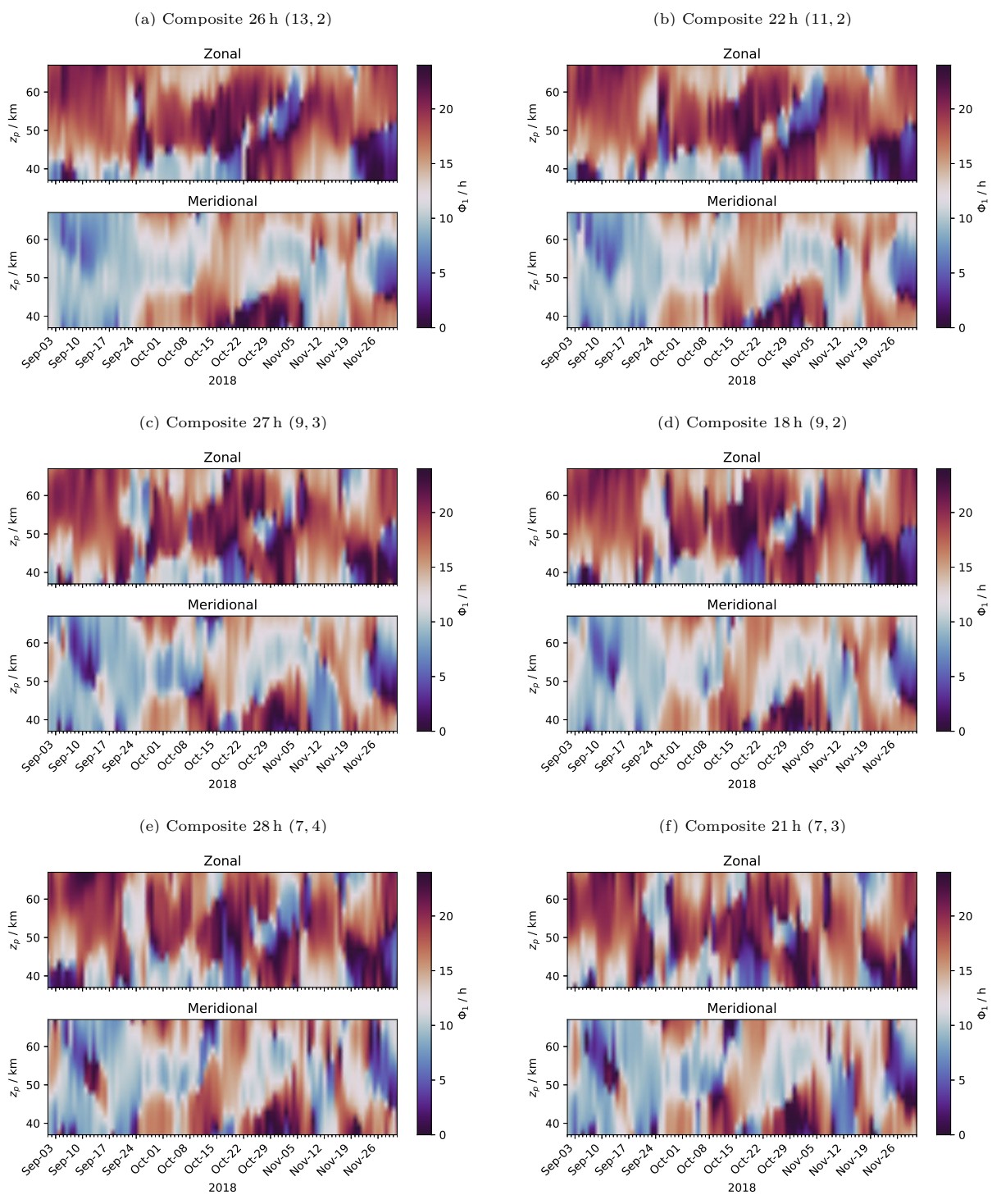

**Figure A4.** Arctic campaign: Phase of the diurnal tide for six different composites with a similar total integration time of 21 h to 28 h ordered by window length.