# Peer review of "First measurements of tides in the stratosphere and lower mesosphere by ground-based Doppler microwave wind radiometry"

_Atmospheric Chemistry and Physics, 2019_

## Referee Comment (RC1) · Anonymous Referee #1 · 11 Oct 2019

General comments

This is an interesting manuscript dealing with the extraction of daily tidal signatures in winds in the stratosphere and lower mesosphere from ground-based Doppler MW measurements. The results are novel, of high interest to the community and the manuscript is in principle suitable for ACP. I do have several comments, some of them not only minor that – in my opinion – should be addressed before the paper is published.

My main concern is, whether the intermittency found in the analysis of the tidal parameters over the 7-day or 13-day time scales is real. As displayed in Fig. 2, the day to day wind variations do not really show an obvious diurnal tidal signature, which suggests

that averaging over extended periods of time is required to suppress the "noise" and to identify the tidal signature with high significance. I would like to point out that I do not question the presence of the tidal signature in the MW data set in general. The analysis of the 3-month periods is quite convincing in my opinion. But the analysis does not rule out the possibility that the relatively strong intermittency in tidal parameters is in fact due to "noise". I make some specific suggestions how to address that and which parts of the paper should be adjusted somewhat. Some of the conclusions drawn are not fully justified in my opinion.

Specific comments

Page 1, line 4: "Current lidar and satellite techniques measure atmospheric tides only in the temperature field and continuous measurements of the tides in the wind field of the stratosphere and lower mesosphere are not available."

This statement is not entirely incorrect, but existing Doppler lidar mesaurements in principle allow studying tides in atmospheric winds, too. These measurements have been used, e.g. to investigate GW signatures in middle atmospheric winds (Baumgarten et al., Geophys. Res. Lett., 42, 10,929 – 10,936, doi:10.1002/2015GL066991). I admit these measurements are not continuous over longer periods of time.

Page 2, line 23: "evident above approximately 40 km altitude and the spread between them is quite large in the lower mesosphere."

It would be good to provide some values here.

Page 2, line 32: "presents" -> "presented" ?

Page 2, line 34: "complimented" -> "complemented"

Page 3, line 28: "the Doppler shift introduced to the emission line is directly proportional to the wind speed"

Only the line of sight wind can be measured, which perhaps should be mentioned. If

zonal and meridional winds are measured, then the radiometer must allow for different viewing directions. I suggest discussing this aspect of the instrumental setup in one or two sentences.

Page 5, lines 4 – 9: the approach used here is essentially a "composite analysis" or "superposed epoch analysis". Perhaps you can mention these terms (or one of them), because the readers will probably be more familiar with these terms than with "aggregation scheme"

Page 6, line 16: "In this study, we do not apply any vertical smoothing to the reanalysis data."

Why not? The reanalysis data could easily be smoted with the radiometer averaging kernels. I suggest doing that. It will affect the results and it can be easily implemented. So, why not doing it?

Page 6, line 19: "We further estimate the uncertainty of the amplitude and phase for the 3-monthly mean using a bootstrapping method by re-sampling the wind time-series"

How is the resampling done? Is it entirely random? There are many different ways to do that and the detailed approach chosen will directly affect the uncertainty estimates and/or the significance of the results. Please provide a brief description, how this was done.

Page 7, line 10 and Figure 1: I think you didn't explain what you mean by "background winds"? Perhaps I missed. It would also be good to mention what the temporal resolution of the background winds is – probably 1 day?

Page 7, line 16 and Figure 2: In my opinion Fig. 2 does not convincingly demonstrate that the measurements capture the diurnal tidal signature. The "noise" (or oscillations, natural variability etc.) in the time series is (are) quite large. Apart from showing the time series over a period of 7 day, I suggest also showing a plot of the superposed tidal signatures, i.e mean diurnal variation averaged over 7 days or an extended period of

time. A composite analysis will suppress the noise and show, whether a tidal signature is visually present in the data. In addition, the superposed tidal signature should also be shown for the 3-month period. I imagine that this plot reveals a remarkably clear tidal signature. This would strengthen the paper significantly in my opinion.

Page 7, line 24: "Further, our observations reveal an intermittency of the diurnal tidal amplitude at the resolved temporal scales of 7 to 13 days"

Looking at Figure 2, I wonder whether this intermittency is real or an artifact. I'm not really convinced all the signatures attributed to the tide are actually caused by it. This intermittency may also be "noise" in the data and not real atmospheric intermittency.

Fig. 3a: Please comment on the vertical structure seen in the amplitude A1. Is this a retrieval artifact? Visually it reminds me of oscillations in the profile retrieval, but it certainly may have other causes.

Page 7, line 28: "is not reproduced in the reanalysis neither in amplitude nor in phase"

Again, the intermittency in the data may not be real.

Page 8, line 4 and Figure 4: red symbols missing in Fig. 4b)

Page 8, line 3: "The green color shows the MERRA-2 reanalysis applying the same temporal aggregation of the time series to derive the tidal amplitudes as for WIRA-C, whereas the red color shows the analysis without averaging."

This description is somewhat different from the description in the caption. The caption mentions smoothed and unsmoothed MERRA data. Perhaps you can use the same description in the text and the Figure caption.

Page 8, line 9: "wavelength of 30 km" -> "wavelength of about 30 km" ?

Page 8, line 10: "In both data sets, the vertical wavelength increases drastically above 55 km altitude, and the phase eventually becomes constant with altitude."

Well, most of the data points are missing for MERRA at altitude above 55 km, so I'm not sure this conclusion is justified for MERRA.

Page 8, line 17: "The shaded area represents the statistical uncertainties of the estimated diurnal tidal amplitudes"

As mentioned above, the way in which the uncertainties are determined should be explained in more detail. There are many possibilities to implement a bootstrapping technique and any uncertainty can be obtained (that's of course a bit exaggerated).

Another question about Fig. 5. The plot shows time series with 1 day resolution. I guess you are showing the results for the 7 day or 13 day analysis at the center day, right? This should be mentioned explicitly and also, whether the 7-day or the 13-day analysis is shown.

Page 9 and Fig. 7: also for the Arctic case discusssed here, the measurements show much more variability and intermittency than the reanalysis. And the agreement between tidal parameters extracted from measurement and model is very good if the full 3 month period is analysed, as shown in Figure 8. This good agreement is probably caused by better "noise" suppression. Again, I suggest performing a composite analysis for the different time periods. The 3-month averaged results seem to be rubust, but I'm not convinced the intermittency is real.

Page 10, line 8: "phase, indicating the presence of highly intermittent diurnal tides."

or just "noise"? Noise should be excluded before drawing the conclusion you drew.

Page 10, line 18: "This is not the case for the measurements and we conclude, that the coherence time of short time scale disturbances is longer in reality than in the reanalysis model."

Again, I think noise should be excluded as a potential explanation before drawing this conclusion.

Figures 4 and 8: please also show uncertainties of the phase values.

It would also be good to show a plot like Figure 2 for the Arctic results.

---

## Referee Comment (RC2) · Anonymous Referee #2 · 18 Oct 2019

The authors describe a novel method of retrieving atmospheric tides from the Doppler microwave wind radiometer observations. The new technique is important for the atmospheric research community due to the limitations of existing ground instruments, lidars and radars, in measuring tides in the range of altitudes from 40 to 70 km. The authors used an original retrieval method for the diurnal tidal components, and compare the retrieved tidal amplitudes and phases with MERRA-2 reanalysis. I believe the manuscript could be published after a revision. I have the following major comments. First, the authors stress the importance of non-zero wind a priori for the retrieval. However, I do not see a clear conclusion of how the non-zero a priori impacts the retrievals. Second, the comparison with MERRA-2 reanalysis requires more details (see comments

below). Third, the authors should comment on the applicability/limitations of radiometer observations for the retrievals of shorter period tides (semidiurnal, etc.) and other oscillations. More specific comments are below.

Line 3: "up to the thermosphere" would be more relevant.

Line 4: "they are gravity waves" is confusing. Perhaps "planetary scale" or "global scale gravity waves", to distinguish from small scale gravity waves?

Line 4: Satellite techniques also measure wind fields associated with tides, though under certain limitations, please clarify.

Lines 9-10: This statement does not reflect the current state of knowledge. Various methods have been utilized to extract short-term variability of tides from satellite observations. A good overview of these methods is given by Ortland, JGR, 2017, doi:10.1002/2016JD025573.

Line 21: RMR lidars can measure winds as well as temperatures in upper stratosphere / lower mesosphere. I assume the authors mean that lidars are not particularly suitable to study tidal oscillations. This needs to be clarified.

Line 3-4: This sentence is out of place and should be (re)moved. The radiometer should be introduced first.

Line 24-33: I believe this method of tidal decomposition has been applied before, also by the authors of this study, e.g., Stober et al., 2017; McCormack et al., 2017. References to the earlier works are needed.

Line 7-9: This needs to be further discussed. Basically this requires some stationarity, both in tidal amplitude and in phase. Perhaps the limitations of retrievals should be also discussed in the Summary section.

Line 31: I am not sure if the chosen interval for Andenes campaign satisfies the proposed criteria. Stronger planetary wave activity starts already in early November.

Line 10-11: I do not I understand the term "composite" here. Do the authors refer to superposed epoch analysis? If they simply refer to complementing WIRA-C with MERRA-2 data, it is better to avoid the "composite" term. More importantly, the authors should detail how the WIRA-C data are complemented with MERRA-2. The representation in Fig. 1 is not clear. When the hatched area goes to higher altitudes (e.g., in meridional winds) – does this correspond to gaps in the WIRA-C dataset?

Line 17 and Fig. 2: The text says "approx. 53 km" but the Fig.2 capture says "approx. 52 km".

Line 31-32 and Fig.3-4: Fig.3 shows, the short term variability is not at all reflected in the reanalysis. From Fig. 4 we can see that the original and smoothed reanalysis show very similar mean behavior (which is not surprising). How would Fig. 3 look if the non-smoothed MERRA-2 reanalysis is analyzed?

Line 14-15 and Fig. 4 a-b: It is surprising to see good mean agreement in zonal component behavior, but poor agreement in meridional above 55 km. Would be useful if the authors add a plot of phase differences between zonal and meridional components as a function of altitude, similar to Fig. 4b, but only the phase differences.

Line 16: Again a mismatch: the text says 55 km altitude and the figure capture says 53

km.

Line 6-11 and Fig.7: Again, the smoothed reanalysis does not reflect the short term variability. How would the amplitudes of non-smoothed reanalysis look?

Line 17-21: Again I would suggest to plot the phase differences as a function of altitude.

---

## Author Comment (AC1) · 19 Dec 2019

December 2019, regarding `https://www.atmos-chem-phys-discuss.net/acp-2019-870`

**Final author comments for acp-2019-870**

**Contents**

**1 Reply to referee comment RC1**

from Jonas Hagen (jonas.hagen@iap.unibe.ch) on behalf of the authors.

Dear Anonymous Referee #1,

we thank you for thoroughly reviewing our manuscript. From your comments, we identify one general comment and a few specific comments, which we address individually below. We are confident, that we can address all comments, in particular those regarding the measurement noise and that the changes we make based on these comments improve the quality and value of our paper.

**General comments**

*Referee1:* This is an interesting manuscript dealing with the extraction of daily tidal signatures in winds in the stratosphere and lower mesosphere from ground-based Doppler MW measurements. The results are novel, of high interest to the community and the manuscript is in principle suitable for ACP. I do have several comments, some of them not only minor that – in my opinion – should be addressed before the paper is published.

My main concern is, whether the intermittency found in the analysis of the tidal parameters over the 7-day or 13-day time scales is real.

As displayed in Fig. 2, the day to day wind variations do not really show an obvious diurnal tidal signature, which suggests that averaging over extended periods of time is required to suppress the "noise" and to identify the tidal signature with high significance. I would like to point out that I do not question the presence of the tidal signature in the MW data set in general. The analysis of the 3-month periods is quite convincing in my opinion. But the analysis does not rule out the possibility that the relatively strong intermittency in tidal parameters is in fact due to "noise". I make some specific suggestions how to address that and which parts of the paper should be adjusted somewhat. Some of the conclusions drawn are not fully justified in my opinion.

*Authors:* We see that the main concern of Anonymous Referee #1 (hereafter AR1) is, that the variability in the composite of our measurement is real and not noise (instrumental or atmospheric).

Our main points that speak in favour of the variability is, (1) that the general morphology of the time dependence of the tidal parameters is consistent among all composites that we looked at and (2) that in the three-monthly mean, the reanalysis and measurement agree very well. No matter if we look at 2 hours aggregated over 13 days or 4 hours over 7 days, we see the same structure in amplitude and phase of the diurnal tide. Nevertheless, some details differ between the composites and that is what we attribute to noise. We include 6 different composits in the revised manuscript's appendix and discuss this method as a mean to test for robustness of our analysis.

That said, we think it is appropriate to somewhat weaken the conclusions. As a consequence, we now first focus on the three-monthly means before we look into the short-time variability. Further, we discuss the influence of noise more thoroughly (also see answers to specific comments).

We thank AR1 for the specific suggestions on how to improve the manuscript and are confident, that we could fully address the general comment.

Resulting Changes:

- Include 6 composites in the appendix.
- Elaborate on robustness of the composites.
- Somewhat weaken the conclusions with regard to variability.
- Change order of figures and discussion of results to focus on mean first.

**Specific comments**

Page 1, line 4: "Current lidar and satellite techniques measure atmospheric tides only in the temperature field and continuous measurements of the tides in the wind field of the stratosphere and lower mesosphere are not available." This statement is not entirely incorrect, but existing Doppler lidar mesaurements in principle allow studying tides in atmospheric winds, too. These measurements have been used, e.g. to investigate GW signatures in middle atmospheric winds (Baumgarten et al., Geophys. Res. Lett., 42, 10,929 – 10,936, doi:10.1002/2015GL066991). I admit these measurements are not continuous over longer

periods of time.

Lidars are in principle capable of measuring atmospheric tides. As AR1 acknowledges, the big difference is the continuity of the measurements. We clarify what we mean by "continuous" and mention the capability of lidars to measure inertia gravity waves.

Commit: Mention that lidars can measure gravity waves.

Resulting Changes:

- Change line in abstract.

- Add sentence: "Current lidar instruments are able to measure inertial gravity waves in the wind field on short timescales (Baumgarten et al., 2015) and are thus in theory also suited for the observation of atmospheric tides, but the necessity of clear sky conditions reduces the availability of long term observations drastically and no observations of atmospheric tides are available to date."

Page 2, line 23: "evident above approximately 40 km altitude and the spread between them is quite large in the lower mesosphere." It would be good to provide some values here.

Thanks, we add the numbers: "Recent findings by Sakazaki et al. (2018) suggest that for the temperature field, differences between the different reanalyses and measurements are systematic in amplitude (approx. 1 K or 50 % above 40 km for northern mid-latitudes, more in tropics) and the spread between the reanalyses is quite large in the lower mesosphere (0.3 K to 1 K at approx. 60 km for northern mid-latitudes). "

Commit: Add numbers from Sakazaki paper.

Page 2, line 32: "presents" → "presented" ?

Page 2, line 34: "complimented" → "complemented"

Thanks for spotting this, we fixed the typos.

Commit: Fix typos.

Page 3, line 28: "the Doppler shift introduced to the emission line is directly proportional to the wind speed" Only the line of sight wind can be measured, which perhaps should be mentioned. If zonal and meridional winds are measured, then the radiometer must allow for different viewing directions. I suggest discussing this aspect of the instrumental setup in one or two sentences.

We add the suggested detail about viewing angles and the projection of the horizontal component to the line-of-sight.

Commit: Clarify observation method and projection of wind speeds.

Resulting Changes:

- Add sentence: "In order to be sensitive to the zonal and meridional component of the horizontal wind speed, we observe the emission line for all cardinal directions (North, East, South, West) at a low elevation angle of 22°."

Page 5, lines 4 – 9: the approach used here is essentially a "composite analysis" or "superposed epoch analysis". Perhaps you can mention these terms (or one of them), because the readers will probably be more familiar with these terms than with "aggregation scheme"

The term *composite analysis* is indeed more common and we choose to use this term, as suggested by AR1.

Commit: Composite analysis vs. aggregation.

Resulting Changes:

- Change "aggregation" to "composite" in most places.

Page 6, line 16: "In this study, we do not apply any vertical smoothing to the reanalysis data." Why not? The reanalysis data could easily be smoothed with the radiometer averaging kernels. I suggest doing that. It will affect the results and it can be easily implemented. So, why not doing it?

There are two aspects we considered in our decision to present the MERRA-2 data without *vertical* smoothing.

Firstly, we use MERRA-2 data on pressure levels (Global Modeling and Assimilation Office (GMAO), 2015) that has already been down-sampled to 42 levels (see Bosilovich et al. (2016) for description of grids). As such, the vertical resolution of the pressure grid above 30 km is approximately 4 km. Additionally, we do apply a temporal smoothing with our composition. As a result, the MERRA-2 data is already smoothed.

Secondly, we see that the biggest uncertainty in the estimate of diurnal tidal amplitude comes from temporal

smoothing by composition and also from instrumental noise. (The effect of vertical smoothing on the phase is negligible.) In this study we provide the comparison with MERRA2 reanalysis data in order to get a reference for amplitude and phase and see what could be expected. The vertical-smoothing error might certainly be subject of further studies.

Page 6, line 19: "We further estimate the uncertainty of the amplitude and phase for the 3-monthly mean using a bootstrapping method by re-sampling the wind time-series" How is the resampling done? Is it entirely random? There are many different ways to do that and the detailed approach chosen will directly affect the uncertainty estimates and/or the significance of the results. Please provide a brief description, how this was done.

Our sampling scheme is inspired by the "Moving Block Bootstrap (MBB)" described by (Lahiri, 2003, "Resampling Methods for Dependent Data", p.25ff).

Detailed description: Assume a $(\Delta D, \Delta H)$ composite, that is $\Delta D$ days composed with a resolution of $\Delta H$ hours (following the nomenclature of the manuscript). So, for each day of the three-month period ($N = 90$ days), we have $\Delta H$-hourly resolved wind speeds that are dependant on the $\Delta D$ surrounding days. Given that data, we construct a synthetic 90 day period (sampling) by choosing $\frac{N}{\Delta D}$ composite days at random (with repetition). By only selecting $\frac{N}{\Delta D}$ composite days, we take into account that the composite makes the data of one day dependent on the surrounding days.

Then, for each sampled 90-day period, we take the mean for each hour of day and fit our tide model to extract the tidal parameters. We estimate the uncertainty of the mean diurnal tidal parameters with the $(0.05, 0.95)$ inter-quantile range. We also check the histogram of the parameters and note that the distribution of the amplitude of the diurnal tide resembles a Gamma distribution (makes sense, because the amplitude cannot drop below zero) and the distribution of the phase resembles a Gaussian distribution. We use the inter-quantile range to represent the distribution in our plots.

As AR1 points out, different resampling methods will result in different uncertainty estimates. We chose the method described above, because it takes the measurement noise into account as well as possible intermittency of atmospheric tides, without assuming anything about the nature of either of them.

Commit: Explain bootstrap.

Resulting Changes:

- Add reference to Lahiri (2003)
- Add short description about the resampling.

Page 7, line 10 and Figure 1: I think you didn't explain what you mean by "background winds"? Perhaps I missed. It would also be good to mention what the temporal resolution of the background winds is – probably 1 day?

Background wind in this study refers to the mean daily wind speed. Yes, the temporal resolution of the background winds as depicted is 24 hours.

Resulting Changes:

- Define what we mean by *background wind* and mention the temporal resolution.

Page 7, line 16 and Figure 2: In my opinion Fig. 2 does not convincingly demonstrate that the measurements capture the diurnal tidal signature. The "noise" (or oscillations, natural variability etc.) in the time series is (are) quite large. Apart from showing the time series over a period of 7 day, I suggest also showing a plot of the superposed tidal signatures, i.e mean diurnal variation averaged over 7 days or an extended period of time. A composite analysis will suppress the noise and show, whether a tidal signature is visually present in the data.

Figure 2 is indeed problematic for different reasons. It depicts a few days in the beginning of one month at one altitude level for one composite and one component only and the relevance of that is questionable. As AR1 suggests, we instead focus on the composite analysis over longer timer periods.

Since this figure lead to some confusion, we decided to remove it from the manuscript, but keep it as Fig. Aa in this discussion. In our opinion it is better to asses the noise from the different composites as we show them in the appendix of the revised paper.

Resulting Changes:

- Remove Figure 2 (keep it as Fig. Aa in this discussion).
- Discuss noise of measurement more extensively.
- Add diurnal cycle for three-monthly mean.

**1 Reply to referee comment RC1**

*In addition, the superposed tidal signature should also be shown for the 3-month period. I imagine that this plot reveals a remarkably clear tidal signature. This would strengthen the paper significantly in my opinion.*

Indeed, these plots strengthen the paper and we add them.

Commit: Add mean level plots.

Resulting Changes:

- Include the figures of the composite tidal signature for the 3 month period.

*Page 7, line 24: "Further, our observations reveal an intermittency of the diurnal tidal amplitude at the resolved temporal scales of 7 to 13 days" Looking at Figure 2, I wonder whether this intermittency is real or an artifact. I'm not really convinced all the signatures attributed to the tide are actually caused by it. This intermittency may also be "noise" in the data and not real atmospheric intermittency.*

The main argument, that speaks in favor of the variability (as opposed to instrumental noise) is that we see the same variability in a large set of different composites. While the composites do not agree in every detail, the general morphology is present in all of them and we would draw the same conclusions no matter which one we look at.

On the other hand, noise is present in our data and we need to discuss the limitations of our method more clearly.

Figure 2 is not meant to demonstrate the capabilities of our method and we see that it is a bit misleading.

Resulting Changes:

- Include 6 composites in the appendix.
- Elaborate on robustness of the composites.

*Fig. 3a: Please comment on the vertical structure seen in the amplitude A1. Is this a retrieval artifact? Visually it reminds me of oscillations in the profile retrieval, but it certainly may have other causes.*

These vertical structures in Fig. 3a (4a in revised manuscript) can also be seen in the reanalysis data (Fig. 4c in revised manuscript), even tough less pronounced and less variable, and also in the 3-monthly mean (Fig. 3a in revised manuscript) in both, measurements and reanalysis data, for the zonal wind with a minimum in amplitude at 55 km.

Since these structures are visible in the extracted amplitude of a composite retrieval, it is not straight forward to relate them to oscillations in the retrieval of wind profiles.

If it has an atmospheric cause, it might be related to mixing of different modes of the diurnal tide (upwards / downwards, trapped / propagating).

Commit: Explain vertical structure of amplitude.

Resulting Changes:

- Discuss vertical structure seen in amplitude: "The vertical structure that is obvious in Fig. 4a can also be seen in the three-monthly mean with a minimum at 55 km in models and measurements. A possible source of this structure is the mixing of different waves with different vertical wavelengths or propagation directions."

*Page 7, line 28: "is not reproduced in the reanalysis neither in amplitude nor in phase" Again, the intermittency in the data may not be real.*

We think it is appropriate to somewhat weaken our conclusions and change this paragraph accordingly.

*Page 8, line 4 and Figure 4: red symbols missing in Fig. 4b)*

Page 8, line 4: this should be orange. Figure 4: They are there but hidden by the green symbols, which is unfortunate.

Resulting Changes:

- We changed the figure to only contain two types of symbols.

*Page 8, line 3: "The green color shows the MERRA-2 reanalysis applying the same temporal aggregation of the time series to derive the tidal amplitudes as for WIRA-C, whereas the red color shows the analysis without averaging." This description is somewhat different from the description in the caption. The caption mentions smoothed and unsmoothed MERRA data. Perhaps you can use the same description in the text and the Figure caption.*

**1 Reply to referee comment RC1**

Thanks, we aligned the caption with the description.

Resulting Changes:

- Align caption with description.

Page 8, line 9: "wavelength of 30 km" → "wavelength of about 30 km" ?

Thanks, we changed that.

Commit: Weaken wavelength.

Page 8, line 10: "In both data sets, the vertical wavelength increases drastically above 55 km altitude, and the phase eventually becomes constant with altitude." Well, most of the data points are missing for MERRA at altitude above 55 km, so I'm not sure this conclusion is justified for MERRA.

We agree and change the sentence accordingly.

Page 8, line 17: "The shaded area represents the statistical uncertainties of the estimated diurnal tidal amplitudes" As mentioned above, the way in which the uncertainties are determined should be explained in more detail. There are many possibilities to implement a bootstrapping technique and any uncertainty can be obtained (that's of course a bit exaggerated).

In this case we use the uncertainty from the error covariance matrix. We changed the sentence accordingly and added an explanation.

Another question about Fig. 5. The plot shows time series with 1 day resolution. I guess you are showing the results for the 7 day or 13 day analysis at the center day, right? This should be mentioned explicitly and also, whether the 7-day or the 13-day analysis is shown.

We use the 13 day aggregate in our manuscript. As mentioned above, we also present the other composites in the appendix and we would draw the same conclusions from every composite we looked at.

Resulting Changes:

- Mention which composite is presented.
- Include reference to the appendix.

Page 9 and Fig. 7: also for the Arctic case discussed here, the measurements show much more variability and intermittency than the reanalysis. And the agreement between tidal parameters extracted from measurement and model is very good if the full 3 month period is analysed, as shown in Figure 8. This good agreement is probably caused by better "noise" suppression. Again, I suggest performing a composite analysis for the different time periods. The 3-month averaged results seem to be robust, but I'm not convinced the intermittency is real.

Also for the arctic campaign, we include now a set of six composites in the appendix. The same time dependence is present in all composites, which we take as a hint that atmospheric variability is at least partly the cause for this time dependence. Anyhow, we agree that some conclusions need to be adjusted and the influence of noise needs to be discussed. (See also response to general comment.)

Page 10, line 8: "phase, indicating the presence of highly intermittent diurnal tides." or just "noise"? Noise should be excluded before drawing the conclusion you drew.

Page 10, line 18: "This is not the case for the measurements and we conclude, that the coherence time of short time scale disturbances is longer in reality than in the reanalysis model." Again, I think noise should be excluded as a potential explanation before drawing this conclusion.

We agree to weaken this conclusions and discuss the presence of noise in more detail.

Figures 4 and 8: please also show uncertainties of the phase values.

We agree.

In the process of re-plotting Fig. 4 and 8, we found that we made a mistake in unwrapping the phase for plotting. This had the effect of separating the WIRA-C and MERRA2 phase for all values of $\Phi > 12$ h.

Resulting Changes:

- Fix the wrong phase unwrapping and error bars.

It would also be good to show a plot like Figure 2 for the Arctic results.

**1 Reply to referee comment RC1**

We added the figure requested by AR1 to this response as Fig. Ab (right next to the original figure 2). We decided to remove Fig. 2 from the new version of the manuscript and added some other figures instead for the tropical and arctic campaign the like.

Best regards,
Jonas Hagen

(a) Tropical (Maïdo, April 2017)

(b) Arctic (ALOMAR, September 2018)

[Figure]

Figure A: Timeseries of retrieved meridional wind from WIRA-C at 50 hPa (approx. 53 km) for beginning of the selected time periods. Additionally, MERRA-2 reanalyses data and the a priori data used for the retrieval are shown. Temporal resolution for WIRA-C and MERRA-2 is 2 hours and 3 hours respectively.

**2 Reply to referee comment RC2**

from Jonas Hagen (jonas.hagen@iap.unibe.ch) on behalf of the authors.

Dear Anonymous Referee #2,

we thank you for thoroughly reviewing our manuscript. From your comments, we identify three general comments and a few specific comments, which we address individually below. We are confident, that we can address all comments, especially those regarding the comparison with the reanalysis and that the changes we make based on these comments improve the quality and value of our paper.

**General comments**

*Referee2:* First, the authors stress the importance of non-zero wind a priori for the retrieval. However, I do not see a clear conclusion of how the non-zero a priori impacts the retrievals.

*Authors:* Since we choose periods with stable background and good measurement response, the selection of the a priori only has a very small impact on the retrieved wind speeds. The same study would be possible with a zero-wind a priori, but we think it would be sub-optimal choice. We do have some arguments to choose a daily-mean a priori over a zero-wind a priori as we will outline below.

For the standard wind retrieval from WIRA-C we use a zero-wind a priori profile. This choice is not obvious, because generally in optimal estimation, one tries to include all a priori knowledge to constrain the optimisation. However, we do this for two main reasons, as described by Hagen et al. (2018): Firstly, wind can change within days from large positive values to large negative values under some circumstances like strong planetary wave activity, sudden stratospheric warmings or sometimes during wind reversals around equinox. The standard approach (zero wind speed a priori and large co-variance) is best suited to cover such cases, which is important to observe the above mentioned effects in year-long time series.

Secondly, it is not very obvious where to get good statistics on dynamics for long-term continuous measurements. There are different candidates, but no matter what one chooses, a big co-variance is necessary to take into account the uncertainty. This is especially the case in the context of the above mentioned disturbances. For a steady background situation, it would perhaps make sense to use some statistics form a climatology and thus reduce the co-variance and thus the uncertainties in the retrieval.

All of the above is relevant to the standard WIRA-C wind retrieval and both reasons for zero-wind a priori profiles have to be revisited for a study on different timescales like the one we present now. We specifically chose periods with no abrupt changes in the wind field (see also specific comment of RC2 about planetary wave activity in polar latitudes). And since we average over 7 to 13 days, we do have high confidence in the ECMWF model data for the mean background wind speed over these time spans. Note, that there is no time-of day dependence of the a priori profile. We retrieve the daily cycle of the wind speed solely from our measurements.

As described in the manuscript, there is something else to consider: Since we want to observe the diurnal cycle of wind speed, we need to make an effort to exclude other sources of diurnal variability from our retrieval. Relevant to the tropical campaign is the troposphere which has pronounced diurnal cycle (water vapour, clouds, precipitation). This means that observations during the after-noon have more noise than observations during night-time, for example. This would, under some circumstances, introduce a daily cycle in the wind speed, because in the after-noon hours, the retrieved wind speed tends more towards the a priori due to increased noise. If the a apriori is zero and the mean background (daily mean) wind speed is around $60\,\mathrm{m\,s^{-1}}$ this effect is hard to quantify and the tendency towards the zero-value would be systematic for after-noon hours in the tropics. This is not necessarily true for the polar latitude, where the weather varies more within days than within a day. With a daily-mean a priori wind profile the amplitude of the measured tide is always decreased by the diurnal cycle of the troposphere, if there is any effect at all.

So, in conclusion, we prefer the daily mean wind speed a priori over the zero wind a priori for the following reasons: It is possible to get a proper background that can be used for the a priori value in the periods and timescales we choose to analyze. The effect of diurnal variability of noise sources on the retrieval of the diurnal cycle is more difficult to characterize for the zero-wind a priori, whereas a daily-mean a priori leads to a smaller amplitude, if there is any effect.

Second, the comparison with MERRA-2 reanalysis requires more details (see comments below).

We improve the comparison with MERRA-2 in the following ways (more details in answers below):

- Add two figures showing the mean diurnal cycle of wind speeds for different altitudes (Fig. 2 and 7

in revised manuscript)

- Add the original (non-smoothed) MERRA-2 data for comparison (Fig. 4e, 4f, 10e and 10f in revised manuscript)

- Add plot of the zonal-to-meridional diurnal wind tide phase difference for measurement and reanalysis (Fig. 9a and 9b)

Commit: Discuss mean diurnal cycle shown in new figures.

Third, the authors should comment on the applicability/limitations of radiometer observations for the retrievals of shorter period tides (semidiurnal, etc.) and other oscillations.

We did not make any studies in that regard. In theory this would be possible with the (13,2,1) or even with a (13,2,0.5) composite that would aggregate by half-time-of-day. Clearly, it is worth mentioning that in the outlook.

Commit: Outlook on semi-diurnal tide.

Resulting Changes:

- Discuss the visibility of higher order tidal modes in Fig. 2 and 7 in revised manuscript.

- Outlook on the possibility to retrieve higher-order tidal modes.

**Specific comments**

Page 1 Line 3: "up to the thermosphere" would be more relevant.

Page 1 Line 4: "they are gravity waves" is confusing. Perhaps "planetary scale" or "global scale gravity waves", to distinguish from small scale gravity waves?

Thanks for these valuable suggestions.

Commit: Thermosphere and planetary-scale

Resulting Changes:

- "ionosphere" → "thermoshpere"

- "gravity waves" → "planetary-scale gravity waves"

Page 1 Line 4: Satellite techniques also measure wind fields associated with tides, though under certain limitations, please clarify.

Page 2 Lines 9-10: This statement does not reflect the current state of knowledge. Various methods have been utilized to extract short-term variability of tides from satellite observations. A good overview of these methods is given by Ortland, JGR, 2017, doi:10.1002/2016JD025573.

Thank you for the hint we changed the manuscript accordingly (see changes).

Resulting Changes:

- Add citation: "The global coverage nevertheless enables tidal studies on shorter timescales also for instruments on these satellites (Ortland, 2017)."

Page 2 Line 21: RMR lidars can measure winds as well as temperatures in upper stratosphere / lower mesosphere. I assume the authors mean that lidars are not particularly suitable to study tidal oscillations. This needs to be clarified.

We clarify the capabilities of lidars with regard to wind and gravity wave measurements.

Commit: Mention that lidars can measure gravity waves.

Resulting Changes:

- Change line in abstract.

- Add sentence: "Current lidar instruments are able to measure inertial gravity waves on short timescales (Baumgarten et al., 2015) and are thus in theory also suited for the observation of atmospheric tides, but the necessity of clear sky conditions reduces the availability of long term observations drastically and no observations of atmospheric tides are available to date."

Page 3 Line 3-4: This sentence is out of place and should be (re)moved. The radiometer should be introduced first.

**2 Reply to referee comment RC2**

We assume that RC2 means Page 3 Line 1-2 and agree that the sentence is out of place. We changed the introduction of microwave radiometry accordingly.

Commit: Rearrange introduction of WIRA-C

Resulting Changes:

- Moved introduction of WIRA-C to the proper paragraph.

Page 5 Line 24-33: I believe this method of tidal decomposition has been applied before, also by the authors of this study, e.g., Stober et al., 2017; McCormack et al., 2017. References to the earlier works are needed.

RC2 suggests to cite earlier work that used a simple harmonic-oscillation model to extract tidal parameters from measurements. We included the suggested references, plus others (see changes). We also included a sentence about the relevant difference to the earlier work.

Resulting Changes:

- Added citations and relation to earlier work in introduction.

Page 6 Line 7-9: This needs to be further discussed. Basically this requires some stationarity, both in tidal amplitude and in phase. Perhaps the limitations of retrievals should be also discussed in the Summary section.

The referenced sentences are: "We assume that the retrieval of averaged spectra yields the average wind speed, so windowing and aggregation can be considered equivalent. This assumption might not hold in the context of fast changes in the wind field and we thus prefer periods of a stable wind background for our detailed analysis."

We agree that the condition for doing composite analysis is stationarity of the observed quantity. This applies to the background as well and thus we perform our analysis only on selected periods with a reasonably stable background.

There is only very little known about the short-time behaviour of tides. We agree, that a stable amplitude and phase is required for a proper composite analysis, but we just do not know this a priori. Based on temperature lidar observations with high temporal resolution, Baumgarten and Stober (2019), for example, provides evidence that tides can indeed be highly intermittent.

As suggested, we discuss this further in the Conclusions section.

Commit: Limitation with regard to stationarity.

Commit: Outlook on non-stationarity.

Resulting Changes:

- Discuss limitations of composite retrievals with regard to stationarity.
- Added outlook regarding requirement of stationary background.

Page 6 Line 31: I am not sure if the chosen interval for Andenes campaign satisfies the proposed criteria. Stronger planetary wave activity starts already in early November.

Planetary waves are indeed active in November 2018 in Andenes as we note in the manuscript (page 8, line 30 of original manuscript): "Both wind components indicate some variability due to waves on temporal scales of a few days, in particular, the meridional wind indicates an onset of the planetary wave activity towards the end of the observation period."

We nevertheless chose this period because in these months, the weather was good (good measurement response), planetary wave activity was low in at least two months and it did not include a wind reversal (that happened just before our period).

We discuss the possible influence of planetary wave activity in the results section.

Page 7 Line 10-11: I do not I understand the term "composite" here. Do the authors refer to superposed epoch analysis? If they simply refer to complementing WIRA-C with MERRA-2 data, it is better to avoid the "composite" term. More importantly, the authors should detail how the WIRA-C data are complemented with MERRA-2. The representation in Fig. 1 is not clear. When the hatched area goes to higher altitudes (e.g., in meridional winds) – does this correspond to gaps in the WIRA-C dataset?

We should not use the term composite in the context of background winds, it is indeed just wrong usage of the word. Fig. 1 in the discussion paper shows a simple overlay, showing WIRA-C data where available (with high quality) and MERRA2 data as visual background. If the hatched area goes higher this does not mean that there is a gap in the WIRA-C data. It just means, that the quality limits were not satisfied for this altitude at that day for the retrieval of daily mean wind speeds.

Commit: Say complemented instead of composite.

Resulting Changes:

- We clarified the description of Fig. 1, it now reads: "The meteorological background wind field for the selected period at the Maïdo observatory is shown in Fig. 1 as measured by WIRA-C, complemented with MERRA-2 reanalysis for lower altitudes.".

Page 7 Line 17 and Fig. 2: The text says "approx. 53 km" but the Fig.2 capture says "approx. 52 km".

Page8 Line 16: Again a mismatch: the text says 55 km altitude and the figure capture says 53 km.

Thanks. We fixed that.

Line 31-32 and Fig.3-4: Fig.3 shows, the short term variability is not at all reflected in the reanalysis. From Fig. 4 we can see that the original and smoothed reanalysis show very similar mean behavior (which is not surprising). How would Fig. 3 look if the non-smoothed MERRA-2 reanalysis is analyzed?

Page 9 Line 6-11 and Fig.7: Again, the smoothed reanalysis does not reflect the short term variability. How would the amplitudes of non-smoothed reanalysis look?

We add amplitude and phase of the non-smoothed MERRA2 data as Fig. 4 and Fig. 10 in revised manuscript.

The non-smooth MERRA2 data shows more tidal variability, as is expected. We have already discussed the non-smoothed data in the context of Fig. 5 and 9 in the original manuscript (Fig. 5 and 11 in revised manuscript), which also show original MERRA2 data and add two paragraphs to discuss the new figures.

Page 8 Line 14-15 and Fig. 4 a-b: Would be useful if the authors add a plot of phase differences between zonal and meridional components as a function of altitude, similar to Fig. 4b, but only the phase differences.

Line 17-21: Again I would suggest to plot the phase differences as a function of altitude.

The phase difference is an interesting quantity to look at, and we already discuss its behaviour in the manuscript, thus it is a good idea to include this plots. We added plots of the phase difference between zonal and meridional wind components for both campaigns as Fig. 9 in revised manuscript.

Commit: Include phase difference plots.

Resulting Changes:

- Include phase difference plots (Fig. 9 in revised manuscript) and references.

Page 8 Line 14-15 and Fig. 4 a-b: It is surprising to see good mean agreement in zonal component behavior, but poor agreement in meridional above 55 km.

Indeed. Assimilation data of the reanalysis model is limited in these altitudes, and this deviation might thus originate from the forecast model itself or boundary conditions imposed on the model. The difference is only visible in the top-most of the 42 pressure levels, so while this is interesting, we can hardly interpret it.

Best regards,
Jonas Hagen

**References**

Baumgarten, G., Fiedler, J., Hildebrand, J., and Lübken, F.-J.: Inertia gravity wave in the stratosphere and mesosphere observed by Doppler wind and temperature lidar, Geophysical Research Letters, 42, 10,929–10,936, https://doi.org/10.1002/2015GL066991, URL `https://agupubs.onlinelibrary.wiley.com/doi/abs/10.1002/2015GL066991`, 2015.

Baumgarten, K. and Stober, G.: On the Evaluation of the Phase Relation between Temperature and Wind Tides Based on Ground-Based Measurements and Reanalysis Data in the Middle Atmosphere, Annales Geophysicae Discussions, pp. 1–29, https://doi.org/10.5194/angeo-2019-25, 2019.

Bosilovich, M., Lucchesi, R., and Suarez, M.: MERRA-2: File specification GMAO Office Note No. 9 (Version 1.1), `http://gmao.gsfc.nasa.gov/pubs/office_notes`, 2016.

Global Modeling and Assimilation Office (GMAO): MERRA-2 inst3_3d_asm_Np: 3d,3-Hourly,Instantaneous,Pressure-Level,Assimilation,Assimilated Meteorological Fields V5.12.4, Goddard Earth Sciences Data and Information Services Center (GES DISC), Greenbelt, MD, USA, https://doi.org/10.5067/QBZ6MG944HW0, 2015.

Hagen, J., Murk, A., Rüfenacht, R., Khaykin, S., Hauchecorne, A., and Kämpfer, N.: WIRA-C: A Compact 142-GHz-Radiometer for Continuous Middle-Atmospheric Wind Measurements, Atmospheric Measurement Techniques Discussions, pp. 1–30, https://doi.org/10.5194/amt-2018-69, 2018.

Lahiri, S.: Resampling Methods for Dependent Data, Springer Series in Statistics, Springer, URL `https://books.google.ch/books?id=e4f8sqm439UC`, 2003.

Ortland, D. A.: Daily estimates of the migrating tide and zonal mean temperature in the mesosphere and lower thermosphere derived from SABER data, Journal of Geophysical Research: Atmospheres, 122, 3754–3785, https://doi.org/10.1002/2016JD025573, URL `https://agupubs.onlinelibrary.wiley.com/doi/abs/10.1002/2016JD025573`, 2017.

Sakazaki, T., Fujiwara, M., and Shiotani, M.: Representation of Solar Tides in the Stratosphere and Lower Mesosphere in State-of-the-Art Reanalyses and in Satellite Observations, Atmospheric Chemistry and Physics, 18, 1437–1456, https://doi.org/10.5194/acp-18-1437-2018, 2018.

---

## Author Response (AR2)

January 2020, regarding `https://www.atmos-chem-phys-discuss.net/acp-2019-870`

**Final author comments for acp-2019-870**

from Jonas Hagen (jonas.hagen@iap.unibe.ch) on behalf of the authors.

Dear Editor, Dear Anonymous Referee #1, Dear Anonymous Referee #2,

Thank you for reviewing the revised manuscript! Based on your comments we made some minor changes to the manuscript.

*Referee1:* Page 2, line 14: "overpass each location on earth at two local times specific to this location": This is a bit misleading, because the measurements from a sun-synchronous platform are all made at the same local time (for a given latitude). So, the local time is not really specific to this location. For latitudes between about 60 S and 60 N the local time of the measurements is within 1 hour of the local time of the equator crossing.

*Authors:* Indeed, this can be written more clearly and sentence now reads:

"Satellites with a sun-synchronous orbit, for example Aura MLS, overpass each location exclusively at two local times which are specific to the latitude of this location."

*Referee1:* Page 2, line 15: "The global coverage nevertheless enables tidal studies on shorter timescales also for instruments on these satellites": This is also a bit misleading, because it's only true for high latitudes, where the satellite "flies" through different local times quickly.

Under certain assumptions (e.g. linear tide model) and with some limitations, tidal studies can be made for equatorial latitudes.

Sentence now reads: "The global coverage nevertheless enables tidal studies on shorter timescales also for instruments on these satellites under the assumption of a linear tide model (Ortland2017)."

*Referee1:* Page 9, line 23: "A possible source of this structure is the mixing of different tidal waves with different vertical wavelengths or propagation directions." CLOSED: [2020-01-22 Mi 14:06] Could this also be a retrieval artifact? The vertical structure reminds me of oscillations often occurring in profile retrievals.

This comment has been raised by AR1 previously. The changes to the manuscript in reply to his/her comment might not have been enough though. We thus elaborate a bit more on the vertical structure in the sense of our reply to AR1's previous comment on this feature.

Paragraph now reads: "Because we observe a similar oscillating pattern of tidal amplitude with altitude in measurements and reanalysis (Fig. 3a), we propose the mixing of different tidal waves with different vertical wavelengths or propagation directions as a possible source for this structure. The augmented magnitude of the structure in our measurements compared to reanalysis data might still be due to slight oscillations of the retrieval."

*Referee1:* Page 12, line 9: "This indicates that the coherence time of short timescale disturbances of the diurnal tide might actually be longer than the reanalysis predicts." It may also indicate that some of the variability seen in the measured data is not real, right? This possibility should also be mentioned/discussed.

Right. The referenced sentence now reads: "If this variability comes from the atmosphere as opposed to instrumental noise, this would indicate that the coherence time of short timescale disturbances of the diurnal tide might actually be longer than the reanalysis predicts."

*Referee1:* Page 5, line 20: ".. we apply the same composition" "Composition" may be misleading here. I suggest using "data treatment", but leave it to the authors to decide.

Page 6, line 2: "compose" Same comment as above. I'm not sure all readers will interpret this correctly.

Thanks for the hint. Indeed, while technically correct, "composition / to compose" is usually not used in this sense. We decided to use the term "aggregation / to aggregate" for the data treatment necessary for a composite analysis.

*Referee1:* Page 1, bottom line: "transportation" → "transport"

Page 2, line 28: "This leaves reanalyses data" → "This leaves reanalysis data"

Page 2, line 34: "become more" → "became more"

Page 4, line 26: "The most important to us is" Something is missing here, i.e. the subject of the sentence.

Page 8, line 28: "westwards observation direction" → "westward observation direction"

Page 9, line 9: "lower most" → "lowermost"

Thank you for spotting these, we fixed the typos.

*Editor:* P8, L13: Figure 9a is referenced after Figure 3. So the order of figures needs to be checked once again.

Thanks, we moved Figure 9 to be Figure 4. Now the order is correct.

*Editor:* P10, L6: one of the closing parentheses is obsolete.

Thanks!

Further, we made a small addition to the acknowledgements to thank the two anonymous referees for their constructive comments.

Best regards,
Jonas Hagen

(Following pages contain a marked-up manuscript version with changes highlighted.)

[revised manuscript text omitted]

---

## Author Response (AR3)

**Final author comments for acp-2019-870**

from Jonas Hagen (jonas.hagen@iap.unibe.ch) on behalf of the authors.

Dear Editor,

Based on your comments we made some minor changes to the manuscript.

*Editor:* there are two minor technical correction that remain to be taken care of before publication. I think your changes concerning the first two comments of referee 2 did not change the fact that the sentences are misleading. Therefore, I have the following two suggestions for rephrasing:

Page 2, L14: Isn't it easier to just write: "Satellites with a sun-synchronous orbit, as for example Aura MLS, overpass each location at two local times a day." or even better "........twice a day". Then you could also add the equator crossing times which are usually given as reference.

*Authors:* The referenced sentence is: "Satellites with a sun-synchronous orbit, for example Aura MLS, overpass each location exclusively at two local times which are specific to the latitude of this location."

Admittedly, this sentence is a bit abstract. In our opinion it gives a very short and concise picture of the sun-synchronous orbits and their relation to tidal studies: If one gathers all measurements from e.g. MLS for one location (area with some extent in practice) they only contain measurements from two local times, that is one from the ascending orbit and one from the descending orbit.

Aura orbits Earth roughly 15 times per 24 hours and whether there are two measurements per day (or more or less) that intersect with the chosen area depends on its extent. For mid-latitudes and an area of roughly 400 km width (approx. ground based radiometer field of view), it is true, that there are two measurements per day. This might be more in polar regions (where orbits are closer together) or less in tropical regions (where orbits are more spread), again depending on the chosen area. However, all measurements no matter how many, always have been taken at one of two local times specific to the latitude of the location (with a small tolerance) and this is relevant for tidal studies.

Thus we would rather not include the suggestion of the Editor and prefer our general statement that is not specific to ground based radiometry and areas of specific size.

*Editor:* Page 2, L15: Here it is not clear to which satellites this sentence refers to. I guess you mean the former ones, the non sun-synchronous ones. Did I understood that correct, that for these you need the linear tide model to accomplish the tidal studies on shorter timescales. In that case you could simply change the sentence as follows: "Nevertheless, the global coverage of the former satellites enables tidal studies on a shorter timescale under the assumption of a linear tidal model (Ortland, 2017).

Yes this is correct. We will swap the sentences so that it is clear that this method only applies to satellites with a general orbit and point out the exception right after that. (See changes.)

Best regards,
Jonas Hagen

(Following pages contain a marked-up manuscript version with changes highlighted. Only changed pages are included.)

[revised manuscript text omitted]